# An Eya1-Notch axis specifies bipotential epibranchial differentiation in mammalian craniofacial morphogenesis

Haoran Zhang[1†], Li Wang[1†], Elaine Yee Man Wong[1], Sze Lan Tsang[1], Pin-Xian Xu[2], Urban Lendahl[3], Mai Har Sham[1]*

[1]School of Biomedical sciences, Li Ka Shing Faculty of Medicine, The University of Hong Kong, Pokfulam, Hong Kong SAR, China; [2]Department of Genetics and Genomic Sciences, Mount Sinai School of Medicine, New York, United States; [3]Department of Cell and Molecular Biology, Karolinska Institutet, Stockholm, Sweden

*For correspondence:
mhsham@hku.hk

[†]These authors contributed equally to this work

Competing interests: The authors declare that no competing interests exist.

**Abstract** Craniofacial morphogenesis requires proper development of pharyngeal arches and epibranchial placodes. We show that the epibranchial placodes, in addition to giving rise to cranial sensory neurons, generate a novel lineage-related non-neuronal cell population for mouse pharyngeal arch development. Eya1 is essential for the development of epibranchial placodes and proximal pharyngeal arches. We identify an Eya1-Notch regulatory axis that specifies both the neuronal and non-neuronal commitment of the epibranchial placode, where Notch acts downstream of Eya1 and promotes the non-neuronal cell fate. Notch is regulated by the threonine phosphatase activity of Eya1. Eya1 dephosphorylates p-threonine-2122 of the Notch1 intracellular domain (Notch1 ICD), which increases the stability of Notch1 ICD and maintains Notch signaling activity in the non-neuronal epibranchial placodal cells. Our data unveil a more complex differentiation program in epibranchial placodes and an important role for the Eya1-Notch axis in craniofacial morphogenesis.
DOI: https://doi.org/10.7554/eLife.30126.001

## Introduction

Craniofacial morphogenesis depends on integration of signals from multiple cell types and is a complex morphogenetic process requiring, among other structures, the genesis of cranial placodes and pharyngeal arches. The cranial placodes are ectodermal thickening structures derived from the pre-placodal primordium, an ectodermal region adjacent to the neural plate (*Baker and Bronner-Fraser, 2001*). The cranial placodes are divided into the anterior (adenohypophyseal, olfactory and lens placodes), intermediate (trigeminal placode) and posterior placodes (otic, lateral line and epibranchial placodes) (*Streit, 2007*; *Saint-Jeannet and Moody, 2014*; *Singh and Groves, 2016*) (*Figure 1A*). The epibranchial placodes comprise the geniculate, petrosal and nodose placodes, which contribute to the cranial nerve VII (facial), IX (glossopharyngeal) and X (vagus), respectively (*D'Amico-Martel and Noden, 1983*). It is believed that the principle function of the epibranchial placodes is to produce viscero-sensory neurons by delaminating cells that undergo neuronal differentiation (*Narayanan and Narayanan, 1980*). In contrast, the olfactory and otic placodes also generate non-neuronal secretory or supporting cells, in addition to delaminating neuronal cells (*Patthey et al., 2014*). The pharyngeal arches (PA) are transient structures formed in the pharyngeal region of the early embryonic head and interactions of the pharyngeal epithelium, the endoderm, neural crest and mesenchyme give rise to facial, head and neck structures during development (*Figure 1A*). A key aspect of PA formation is the segmentation process, during which the pharyngeal endoderm

extends laterally towards the ectoderm, forming an epithelial pocket called the pharyngeal pouch, while the surface ectoderm invaginates and forms the pharyngeal cleft (*Graham, 2003*). The contact between ectodermal cleft and endodermal pouch forms the segmental plate, which gradually extends proximal-distally to define the boundary between each PA, and restricts the route for neural crest migration into each arch (*Graham et al., 2005*; *Shone and Graham, 2014*). The segmental contact points are also sources of signaling factors that regulate the anterior-posterior and proximal-distal patterning of the PA and promote mesenchymal cell survival (*Trumpp et al., 1999*; *Couly et al., 2002*; *David et al., 2002*). The relationship between epibranchial placodes and PA development is poorly understood, but the notion that the geniculate, petrosal and nodose placodes are located near the pharyngeal clefts (*Figure 1A*), suggests that they are part of the coordinated development of the corresponding PA.

Eya and Six transcription factors are highly evolutionarily conserved and among the first factors expressed across the pre-placodal region where they are critical regulators of placodal cell differentiation in later stages (*Kozlowski et al., 2005*; *Chen et al., 1997*; *Pandur and Moody, 2000*; *Christophorou et al., 2009*; *Zou et al., 2004*; *Saint-Jeannet and Moody, 2014*). Six proteins are DNA-binding activator proteins that promote expression of pre-placodal genes when partnered with Eya. In contrast, Eya does not bind DNA directly, but acts as the transactivating partner to Six proteins. The importance of Eya and Six proteins for craniofacial development is underscored by that mutations in the human *EYA1* and *SIX* genes, including *SIX1* and *SIX5,* are reported in approximately 50% of the patients suffering from Branchio-Oto-Renal (BOR) syndrome (*Abdelhak et al., 1997*; *Smith, 1993*). *Eya1* mutant mouse embryos display phenotypes in multiple pharyngeal and placodal derivatives including cranial neural crest cell-derived bones and cartilages, endoderm-derived tympanic cavity, thymus, thyroid and parathyroid glands, ectoderm-derived external auditory canal, otic and epibranchial placodes (*Xu et al., 1999*, *2002*; *Zou et al., 2004*, *2006*). $Six1^{-/-}$ and $Six4^{-/-}$ mouse embryos also exhibit pharyngeal phenotypes along with kidney problems (*Laclef et al., 2003*; *Zou et al., 2006*), in keeping with a tight functional coupling between Eya and Six proteins. However, the Eya1 protein does not only serve as a transcriptional co-activator, but also possesses tyrosine and threonine phosphatase activities (*Li et al., 2003*; *Okabe et al., 2009*; *Tadjuidje and Hegde, 2013*). The tyrosine phosphatase activity of Eya1 regulates H2AX and DNA repair (*Cook et al., 2009*) and is also required for the function of Eya1 as a transactivator in conjunction with Six (*Li et al., 2003*). Less is known about the threonine phosphatase activity, but dephosphorylation of T58 in c-Myc by Eya1 has been shown to be important for nephrogenesis and tumour growth (*Xu et al., 2014*; *Li et al., 2017*).

Craniofacial development is regulated by Notch signalling. Genetic and pharmacological inhibition of Notch results in proximal PA and craniofacial defects as well as premature neurogenesis and neuronal delamination (*Humphreys et al., 2012*; *Zuniga et al., 2010*; *Daudet and Lewis, 2005*; *de la Pompa et al., 1997*; *Lassiter et al., 2010*; *Lassiter et al., 2014*). Conversely, activation of Notch in the cranial placodes prevents both neurogenesis and delamination and expands the placodal epithelium (*Jayasena et al., 2008*; *Pan et al., 2013*). The Notch signalling pathway is an evolutionarily ancient cell-cell communication system that controls differentiation in most tissues and organs. Upon activation by ligand on a juxtaposed cell, the Notch receptor is proteolytically processed and the final cleavage, which is executed by the γ-secretase complex, liberates the C-terminal portion of the Notch receptor, the Notch intracellular domain (Notch ICD). Notch ICD is translocated into the nucleus to form a complex with the DNA-binding protein CSL, which converts CSL from a repressor to an activator, leading to activation of Notch downstream genes, including *Hes* and *Hey* genes (*Siebel and Lendahl, 2017*). Notch gene dosage and the level of Notch receptor mediated signaling have profound impact on cell fate decision in many cell types during development (*Andersson and Lendahl, 2014*).

Here we address novel roles of Eya1 in regulating PA and epibranchial placode differentiation. We show that Eya1 is required for proper proximal PA, pharyngeal cleft and pouch formation. We also provide evidence that epibranchial placodes are not limited to generating viscero-sensory neurons only, but contain bipotential epithelial progenitors that in addition produce a non-neuronal cell lineage important for PA development. We further identified Notch1 ICD as a novel target of the threonine phosphatase activity of Eya1. Dephosphorylation leads to stabilization of Notch1 ICD and enhanced Notch signaling. Our data provide evidence for a more complex composition of epibranchial placodes and an Eya1-Notch axis important for PA formation and craniofacial morphogenesis.

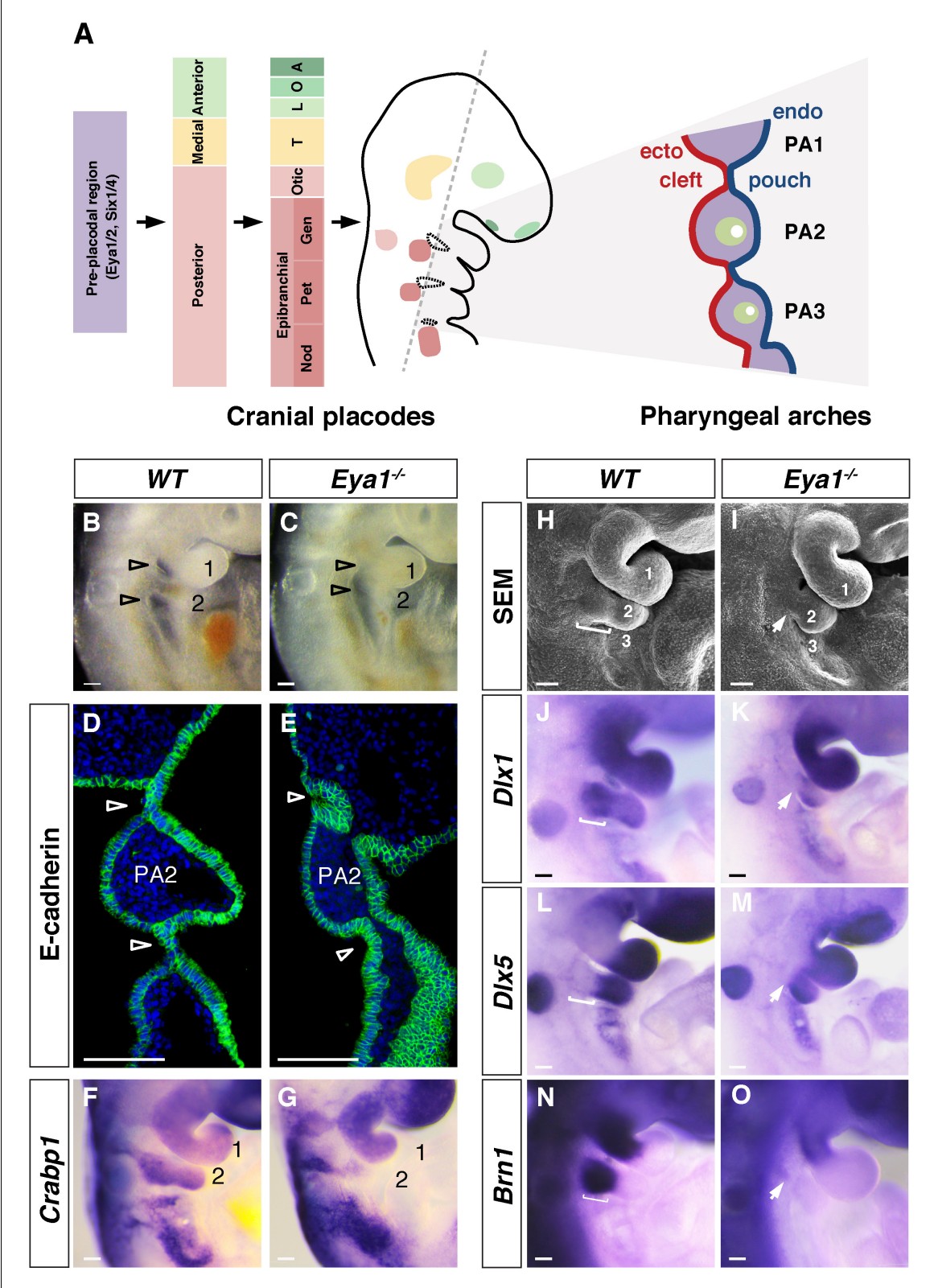

**Figure 1.** Cranial placode and pharyngeal arch development in wildtype and *Eya1*^-/- mouse embryos. (**A**) Schematic summary of the development of cranial placodes and pharyngeal arches (PA) in mouse embryos. The pre-placodal region, marked by expression of *Eya* and *Six* families of genes at E8.0, is divided into anterior, medial and posterior placodal regions at E8.5, which further develops into specific cranial placodes (A, adenohypophyseal; O, olfactory; L, lens; T, trigeminal; Gen, geniculate; Pet, petrosal; Nod, nodose) from E8.5–9.5. The epibranchial placodes are located in close proximity

*Figure 1 continued on next page*

*Figure 1 continued*

to the pharyngeal segmental plates (circled with black dotted lines). The grey dashed line indicates the plane of coronal section, which reveals the pharyngeal segmental plates and arch structures as shown in the diagram on the right (also panel D and E). The PA structures include the pharyngeal ectoderm (red), endoderm (blue) and the transient pharyngeal segmental plates, which form the clefts and pouches. The neural crest, mesoderm and aortic arch arteries are indicated in purple, green and white, respectively. (**B and C**) Lateral view of wildtype (*WT*) and *Eya1*-/- whole mount E9.5 embryos. Open arrowheads indicate positions of pharyngeal clefts; PA1 and PA2 are numbered (n > 20). (**D and E**) Immunostaining for E-cadherin (green) and DAPI (blue) on coronal sections of *WT* and *Eya1*-/- E9.5 embryos. Arrowheads indicate positions of the pharyngeal segmental plates, which are not formed in *Eya1*-/- embryos (n = 4). (**F and G**) Whole-mount in situ hybridization showing *Crabp1* expression in *WT* and *Eya1*-/- E9.5 embryos (n = 5). (**H and I**) Scanning electron microscopy images of *WT* and *Eya1*-/- embryos at E10. White bracket in WT embryo indicates the proximal region of PA2, which was missing in *Eya1*-/- embryos (indicated by arrow) (n = 5). (**J–O**) Expression of *Dlx1*, *Dlx5* and *Brn1* in *WT* and *Eya1*-/- E9.5 embryos. White brackets indicate the proximal region of PA2 in *WT* embryos. Arrows indicate the missing proximal PA2 in *Eya1*-/- embryos (n > 5). Scale bars, 100 μm.

DOI: https://doi.org/10.7554/eLife.30126.002

The following source data and figure supplement are available for figure 1:

**Source data 1.** Source data relating to *Figure 1—figure supplement 1E*.
DOI: https://doi.org/10.7554/eLife.30126.004
**Figure supplement 1.** TUNEL assay in WT and *Eya1*-/- E9.5 embryos.
DOI: https://doi.org/10.7554/eLife.30126.003

## Results

### *Eya1*-/- embryos exhibit defective pharyngeal arch segmentation and lack proximal arch structures

The first PA became morphologically identifiable at E8.5, at this stage wildtype and mutant PA appeared grossly comparable in size and morphology (data not shown). In contrast, at E9.5, *Eya1*-/- embryos showed hypoplastic second and third PA with barely visible pharyngeal clefts (*Figure 1B and C*). E-cadherin immunostaining, which labeled the ectodermal and endodermal epithelia, revealed that the pharyngeal segmental plates were not formed. There were no contact points between the pharyngeal ectoderm and endoderm-derived epithelia, such that the pharyngeal clefts and pouches could not form properly and the pharyngeal mesenchyme in adjacent PA never separated (*Figure 1D and E*).

In E10 *Eya1*-/- embryos the PA2 was shortened, with an apparent reduction of the proximal portion, while the distal part was still present (*Figure 1H and I*). To assess this further, we performed in situ hybridization with three markers that have distinct expression patterns along the proximal-distal axis: *Dlx1* (expressed in both the proximal and distal parts), *Dlx5* (expressed only in the distal part) and *Brn1* (Pou3f3; expressed only in the proximal part) (*Jeong et al., 2008*). At E9.5, *Dlx1* expression was lost in the *Eya1*-/- embryos in the proximal portion of the PA, but retained in the distal arches (*Figure 1J and K*). *Dlx5* was expressed in the distal region in both wildtype and *Eya1*-/- embryos (*Figure 1L and M*), whereas *Brn1* expression was lost in proximal PA2 in *Eya1*-/- embryos but expressed in the maxillary-mandibular junction of PA1 and proximal region of PA2 in wildtype (*Figure 1N and O*). We examined the expression of the neural crest marker gene *Crabp1* and confirmed that the migrating neural crest stream in the PA2 was reduced in the *Eya1*-/- embryos (*Figure 1F and G*). TUNEL analysis revealed that the *Eya1*-/- embryos displayed increased cell death at the dorsal level around the entrances of PA1 and PA2 (*Figure 1—figure supplement 1A–1E*). Taken together, the analysis of the *Eya1*-/- phenotype shows that the abnormal pharyngeal arch segmentation and proximal PA defects are likely the basis for the abnormal morphogenesis in *Eya1*-/- embryos previously described (*Xu et al., 1999*).

### *Eya1*-/- specific phenotypes in the pharyngeal arch and epibranchial placode

*Eya1* and *Six1* are both important for craniofacial morphogenesis (*Xu et al., 1999*; *Laclef et al., 2003*; *Zou et al., 2006*) and we first compared their expression patterns. At E8.5 and before the pharyngeal arch outgrowth, *Eya1* was expressed broadly in the posterior placode and the pharyngeal region, including the pharyngeal ectoderm and pharyngeal pouch (*Figure 2A and A'*). At E9.5, *Eya1* was restricted to the maxillary and most distal part of PA1, the proximal-rostral and caudal parts of PA2, and the posterior PA (*Figure 2B and B'*). Eya1 immunoreactivity was observed in both

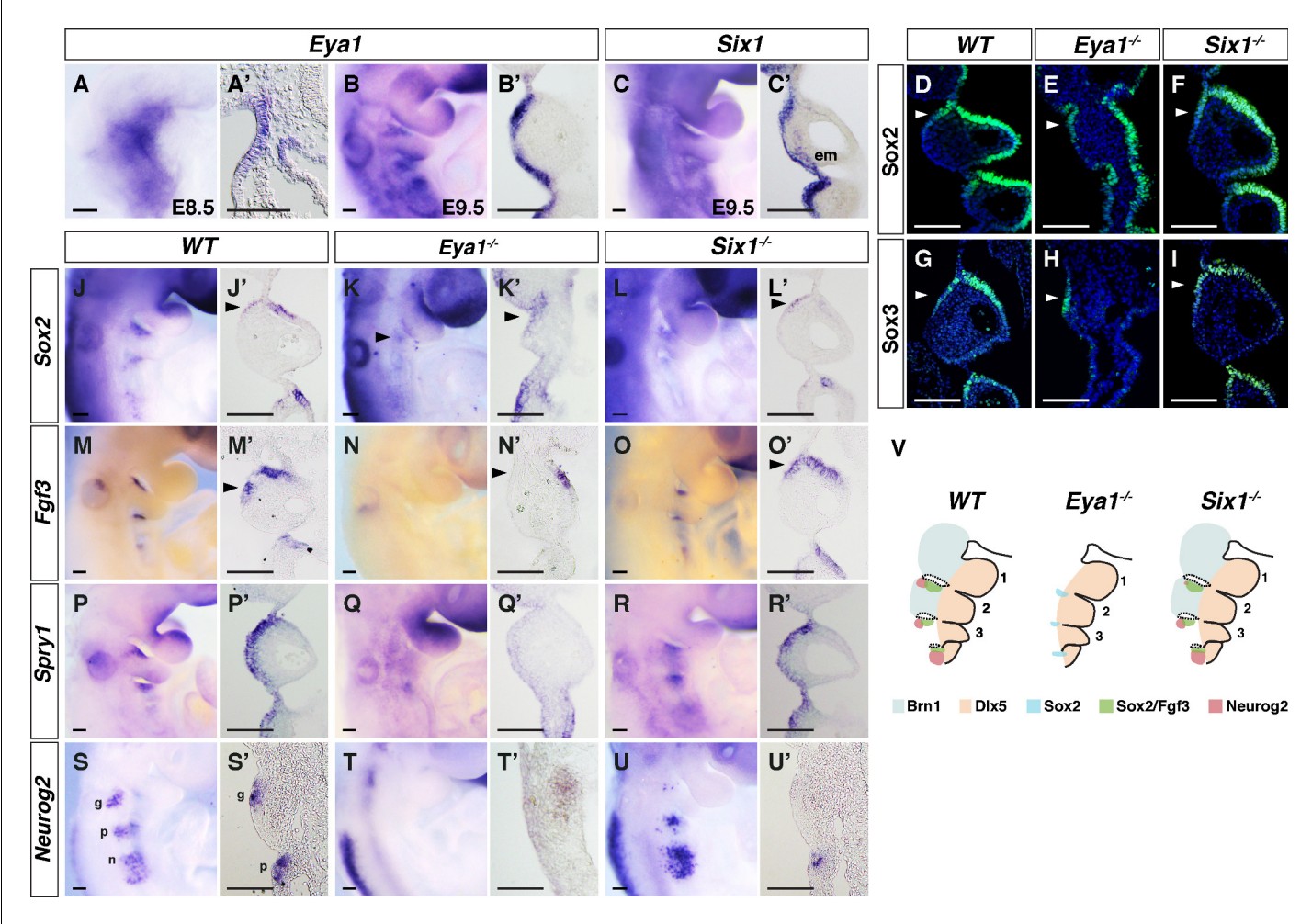

**Figure 2.** Differences in pharyngeal arch phenotypes in *Eya1⁻/⁻ and Six1⁻/⁻* embryos. (A–C′) In situ hybridization on whole-mount and coronal sections of *Eya1* and *Six1* in *WT* embryos at E8.5 (n = 5) and E9.5 (n = 5). (D–I) Immunostaining for Sox2 and Sox3 on coronal sections of E9.5 *WT, Eya1⁻/⁻* and *Six1⁻/⁻* embryos. (J–R′) In situ hybridization of *Sox2, Fgf3,* and *Spry1* on whole-mount and coronal sections of E9.5 embryos (n ≥ 5). (S–U′) In situ hybridization of *Neurog2* on whole-mount and coronal sections through the region of the geniculate and petrosal placodes of E9.5 embryos (n = 5). (V) Schematic summary of the abnormal PA phenotype and gene expression patterns in *WT, Eya1⁻/⁻* and *Six1⁻/⁻* embryos. Light blue and light orange label the *Brn1⁺* proximal and *Dlx5⁺* distal PA regions, respectively. The black dotted circles demarcate the pharyngeal segmentation plates. Green, blue, and red indicate *Sox2⁺* and *Fgf3⁺*, single *Sox2⁺* and *Neurog2⁺* expression regions, respectively. em, ectomesenchyme; g, geniculate placode; p, petrosal placode; n, nodose placodes. Arrowheads indicate rostral-proximal ectodermal cells of PA2. Scale bars, 100 µm.

DOI: https://doi.org/10.7554/eLife.30126.005

The following figure supplement is available for figure 2:

**Figure supplement 1.** Expression of Eya1, *Fgf8, Fgf15,* and *Fgfr1* in the pharyngeal ectoderm of E9.5 embryos.

DOI: https://doi.org/10.7554/eLife.30126.006

cytoplasm and nucleus in ectodermal cells (*Figure 2—figure supplement 1A and A′*). The expression pattern of *Six1* was similar to *Eya1* at E9.5 (*Figure 2C*), although *Six1* expression also extended into the pharyngeal ectomesenchyme (*Figure 2C′*).

*Sox3⁻/⁻* and *Fgfr1ⁿ⁷/ⁿ⁷* mutants display similar proximal pharyngeal arch phenotypes (*Trokovic et al., 2003*; *Rizzoti and Lovell-Badge, 2007*) and we therefore analyzed the expression of SoxB1 genes and Fgf signaling factors in *Eya1⁻/⁻* and *Six1⁻/⁻* E9.5 embryos. Analysis of *Sox2*, which is an early marker for cranial placodes (*Wood and Episkopou, 1999*), revealed expression in the rostral-proximal part of the pharyngeal ectoderm (*Figure 2J and J′*) and a reduction of *Sox2* expression in both *Six1* and *Eya1* knockout embryos (*Figure 2K and L*). However, Sox2⁺ cells were still observed in the pharyngeal ectoderm in both mutants (*Figure 2D–2F*). In *Eya1⁻/⁻* embryos, Sox2⁺

cells were found in the residual proximal pharyngeal ectoderm, and the total number of Sox2$^+$ cells in the mutant geniculate placode (139 ± 33, n = 3) was significantly reduced compared with wildtype (286 ± 26, n = 3). Expression of *Sox3*, which also regulates pharyngeal segmentation and proximal PA2 development (*Rizzoti and Lovell-Badge, 2007*), was reduced in *Eya1$^{-/-}$* (*Figure 2H*) but unaffected in *Six1$^{-/-}$* ectoderm (*Figure 2I*).

In wildtype and *Six1$^{-/-}$* E9.5 embryos, *Fgf3* expression was observed in the rostral-proximal pharyngeal ectoderm and endoderm (*Figure 2M, M, O and O'*). While *Fgf3* expression in the pharyngeal endoderm was retained in the *Eya1$^{-/-}$* embryos (*Figure 2N'*), *Fgf3* expression was dramatically reduced in the *Sox2$^+$* epithelial cells (*Figure 2N and N'*). *Fgf3* expression stems from *Sox2$^+$* cells (see below), and the fact that these cells were still present in the *Eya1$^{-/-}$* embryos (*Figure 2E*) indicates that the reduction of *Fgf3* expression is not due to the loss of the Fgf3-generating cells. Expression of *Fgf15* and *Fgf8* was also reduced in the *Eya1$^{-/-}$* embryos, but unaffected in the *Six1$^{-/-}$* embryos (*Figure 2—figure supplement 1B–1G*). In contrast, the expression of *Fgfr1* was neither affected in the *Eya1$^{-/-}$* nor in the *Six1$^{-/-}$* embryos (*Figure 2—figure supplement 1H–1J*). We also examined the expression of *Spry1* (*Sprouty1*), a downstream effector of Fgf signaling (*Minowada et al., 1999*). *Spry1* expression in the ectoderm was specifically downregulated in *Eya1$^{-/-}$* embryos but not in *Six1$^{-/-}$* embryos (*Figure 2P–2R'*), indicating a reduced level of ectodermal Fgf signaling activity in *Eya1$^{-/-}$* embryos. This is likely to stem from the Fgf ligand side, as the level of *Fgfr1* was not affected in the mutant embryos (*Figure 2—figure supplement 1H–1J*). Pharyngeal arch segmentation occurred normally in the *Six1$^{-/-}$* embryos (*Figure 2F, I, L', O' and and R'*), whereas it was severely compromised in the *Eya1$^{-/-}$* embryos as described in *Figure 1*. Finally, epibranchial placodal cells in the *Eya1$^{-/-}$* embryos did not express *Neurog2* (*Figure 2T and T'*), while *Neurog2* expression was still observed in the petrosal and nodose placodes in the *Six1$^{-/-}$* embryos (*Figure 2U and U'*). This result verified previous findings that epibranchial placode neurogenesis was affected in both mutants, but more severely in the *Eya1$^{-/-}$* embryos (*Zou et al., 2004*).

In sum, the *Eya1$^{-/-}$* and *Six1$^{-/-}$* embryos differ in some important regards (*Figure 2V*). In the *Eya1$^{-/-}$* embryos, there are pharyngeal arch segmentation defects, loss of proximal arches accompanied by loss of ectodermal *Fgf3* and *Neurog2* expression. Conversely, in the *Six1$^{-/-}$* embryos, the *Sox2$^+$* cells are specified and patterned in the right position and able to produce near normal levels of Fgf3. While it is possible that other *Six* genes such as *Six2/4* may compensate for the absence of *Six1*, collectively, the data suggest that *Eya1* is specifically required to regulate the formation of PA and epibranchial placodes.

## Bipotential epibranchial placode differentiation

To gain insights into the cellular origin of the *Eya1$^{-/-}$* specific phenotypes, we analyzed *Sox2*, *Fgf3* and *Neurog2* expression during epibranchial placode and PA development. The expression patterns and morphological transitions are summarized in *Figure 3* and *Figure 3—figure supplement 1*. To obtain a more high-resolution view of the gene expression patterns, we examined the expression of Eya1, Sox2, Neurog2 and Fgf3 on transverse sections from E8.0 (5 somites) to E9.5 (25 somites) wildtype embryos at the position of the epibranchial or geniculate placodes. At E8.0 (5ss), the thickened placodal ectoderm was Sox2$^+$ and none of the cells was Fgf3$^+$ or Neurog2$^+$ (*Figure 3A and F–G'*). At E8.5 (8ss), Neurog2 and Fgf3 expression started to emerge in the Sox2$^+$ placodal ectoderm (*Figure 3B and I–J'*). Co-staining for Sox2 and Neurog2 showed a salt-and-pepper expression distribution at 15ss (*Figure 3L*) and three types of expression patterns could be discerned: Sox2 single positive, Neurog2 single positive, and Sox2/Neurog2 double positive cells. *Fgf3* expression could also be detected in the Sox2$^+$ cell population (*Figure 3C and M–M'*). At 8-15ss, there were Fgf3/Sox2 double positive cells in the Neurog2 positive region (*Figure 3I–3J' and and L–M'*). Interestingly, at E9.5 (25ss), all Neurog2$^+$ cells were sorted to the proximal region while the Sox2$^+$/Fgf3$^+$ cells were confined to the more distal region on the lateral surface of the PA (*Figure 3O–3P'*). Eya1 was co-expressed with Sox2 from 5ss onwards and was expressed in both the Neurog2$^+$ and Fgf3$^+$ regions from 8 to 25ss (*Figure 3E, H, K and N*). Since the Fgf3$^+$/Sox2$^+$/Neurog2$^-$ cells also exhibited a thickened epithelial morphology and expressed *Eya1*, we propose that these cells define a new non-neuronal cell lineage in the epibranchial placode. In *Eya1$^{-/-}$* embryos, Neurog2 expression was lost while Sox2 expression was still retained in the epithelium from E8.0 to E9.5 (*Figure 3—figure supplement 1A–1D'*). Therefore, while the Neurog2$^+$ neurogenic placodal cells were lost, the Sox2$^+$

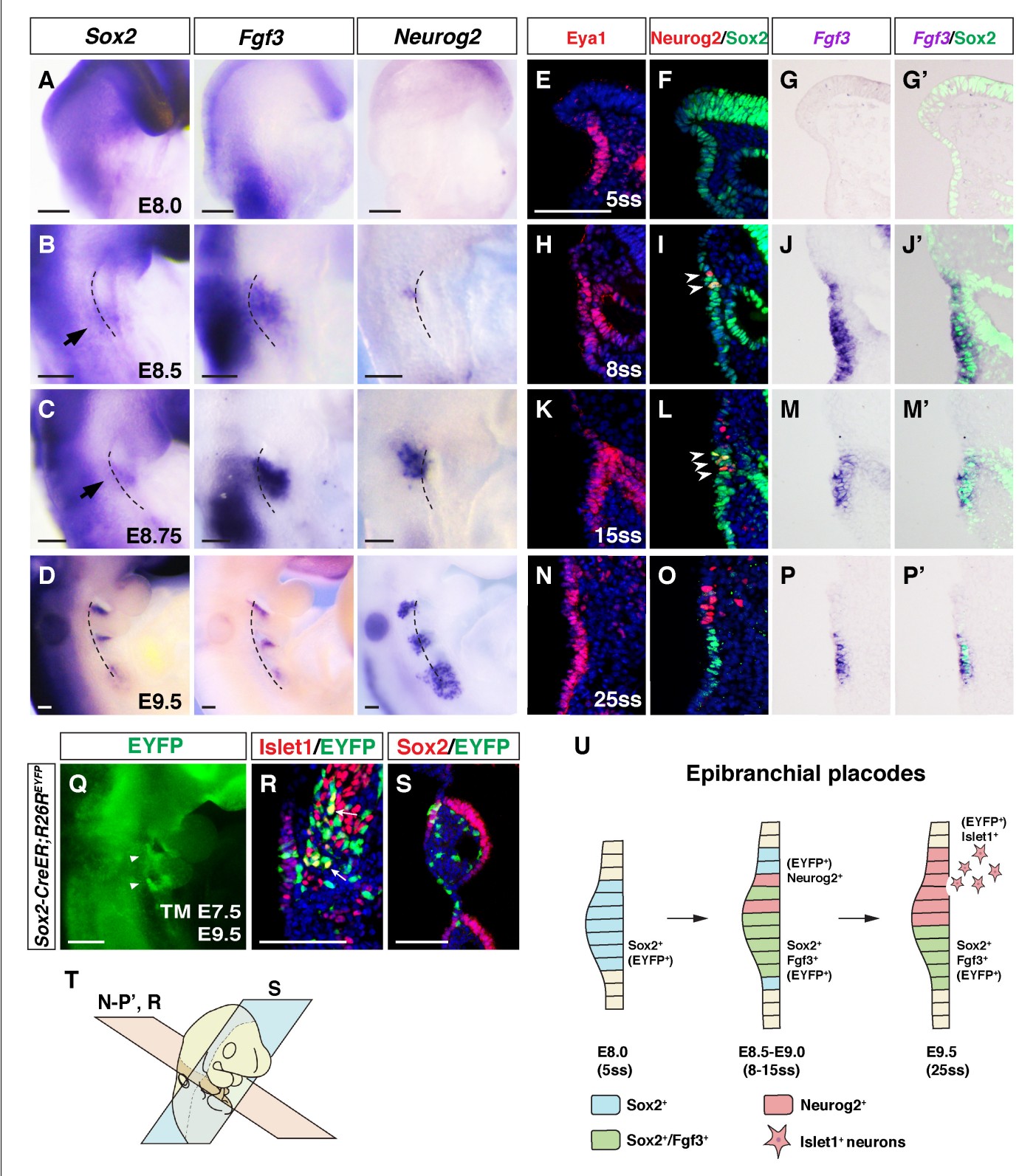

**Figure 3.** Bipotential Sox2⁺ progenitors give rise to both neurogenic and non-neuronal epibranchial placodal cells. (A–D) *Sox2*, *Fgf3* and *Neurog2* expression in whole-mount *WT* embryos at E8.0, E8.5, E8.75 and E9.5. At E8.0, the pharyngeal surface ectoderm was induced to form the thickened posterior placode, which expressed Sox2 but not *Fgf3* or *Neurog2* (A). At E8.5, the first PA became morphologically identifiable and the pharyngeal ectoderm began to make contact with the endoderm. *Fgf3* and *Neurog2* expression could be detected in the epibranchial placode (B). At E8.75, the

*Figure 3 continued on next page*

*Figure 3 continued*

first pharyngeal cleft was already visible on the lateral side of the embryo, just rostral to the group of *Sox2*⁺cells, and *Fgf3* and *Neurog2* expression expanded ventrally and dorsally, respectively (**C**). At E9.5, the *Sox2* and *Fgf3* signals became more condensed and restricted to the rostral-proximal PA, while the *Neurog2* expression expanded more dorsally in each PA (**D**), and the separation of the *Sox2*⁺/*Fgf3*⁺ and *Neurog2*⁺ cells became clear. Arrows indicate *Sox2*-expressing epibranchial placodal regions (n ≥ 5 for each stage). Black dashed lines indicate the position of foregut dorsal endoderm. (**E–P'**) Immunostaining for Eya1 (**E, H, K and N**) and co-immunostaining of Neurog2 and Sox2 (**F, I, L and O**) on adjacent transverse sections; and co-staining of Sox2 and in situ hybridization of *Fgf3* (**G, J, M and P**) on transverse sections of *WT* embryos at different somite stages (ss) as indicated. Arrowheads indicate Neurog2⁺/Sox2⁺ double positive cells (n ≥ 4 per stage). (**Q–S**) Lineage tracing of Sox2⁺ placodal cells in *Sox2-Cre; R26R^EYFP^* embryos. Tamoxifen was injected at E7.5 and embryos examined by whole-mount EYFP fluorescence at E9.5 (n = 10) (arrowheads indicate EYFP⁺ placodal regions) (**Q**). Co-immunostaining for Islet1/EYFP and Sox2/EYFP revealed the neurogenic lineage (n = 6) (**R**) and non-neuronal lineages (n = 4) (**S**). Arrow indicates Islet1⁺/EYFP⁺ positive cell. (**T**) Schematic diagram illustrating the coronal (blue; **S**) and transverse (pink; **E–P' and R**) planes of sections for the respective panels in this figure. (**U**) Schematic diagram summarizing the differentiation of Sox2⁺ epibranchial placode progenitors (blue) into Neurog2⁺ neurogenic placodal cells and delaminating neurons (red); and Sox2⁺/Fgf3⁺ placodal cells (green). Scale bars, 100 μm.
DOI: https://doi.org/10.7554/eLife.30126.007

The following figure supplement is available for figure 3:

**Figure supplement 1.** Analysis of bipotential Sox2⁺ progenitors of the neurogenic and non-neuronal lineages in the epibranchial placodes.
DOI: https://doi.org/10.7554/eLife.30126.008

cells in the non-neuronal lineage remained in the *Eya1*⁻/⁻ embryos (depicted in **Figure 3—figure supplement 1E**).

We next asked whether the Neurog2⁺ cells that delaminate to form neurons in the cranial ganglia and the Fgf3⁺/Sox2⁺/Neurog2⁻ cells were lineage-related. To address this, we used a Sox2 cell lineage tracing system (*Sox2-CreER; R26R^EYFP^* mice) in which tamoxifen induction at E7.5 would label the Sox2-expressing cells after this stage. Analysis of tamoxifen-injected embryos two days later, at E9.5 (**Figure 3Q**), revealed that some EYFP-expressing cells were also positive for Islet1, a marker for delaminating neurons (**Figure 3R**), suggesting that the Sox2⁺ lineage gave rise to neuronal progeny destined for the geniculate ganglion. We however also observed EYFP-expression in cells with thickened epithelial morphology, which expressed Sox2 and also gave rise to the pharyngeal cleft, and proximal pharyngeal ectoderm cells (**Figure 3—figure supplement 3S**). This indicates that cells expressing Sox2 at E7.5 segregated to delaminating neurons as well as to cells in the non-neurogenic lineage. In conclusion, these data provide evidence for a novel type of lineage that 'lives on' in the epibranchial placode after delamination of neuronal cells, and continues to express Sox2, Fgf3 and other factors. By injection of tamoxifen at E9.5 and analysis at E12.5 and E14.5, we detected EYFP expression in the external auditory canal and pinna epithelium, indicating that non-neuronal cells were produced (**Figure 3—figure supplement 1H and I**). The segregation of Sox2, Neurog2 and Fgf3 expression from E8.0–9.5 leading to the genesis of delaminating neuroblasts and the non-neuronal lineage are schematically depicted in **Figure 3U**.

## Notch signaling is reduced in the pharyngeal epithelium of *Eya1*⁻/⁻ embryos

The Notch signaling pathway is a key regulator of the neurogenesis of cranial placodes and cell fate specification (*Hartman et al., 2010*; *Lassiter et al., 2014*). We first analyzed the distribution of components of the Notch pathway and downstream target genes in wildtype embryos. The pharyngeal ectoderm showed extensive co-expression of Notch1 with Eya1 (**Figure 4—figure supplement 1A–1A''**) and Sox2 from E8.0 onwards (**Figure 4—figure supplement 1O and Q**). Notch immunoreactivity was observed also in the nuclei, indicating active Notch signaling, a notion further supported by expression of Hey1, a Notch downstream target, at the pharyngeal ectoderm, with a distribution similar to that of Sox2 from E8.5 to E9.5 (**Figure 4—figure supplement 1B–1E**). In contrast, the expression domain of the Notch ligand Delta-like1 (Dll1) was found to overlap with that of Neurog2 from E8.5 to E9.5 (**Figure 4—figure supplement 1F–1I**). The expression domain of the Jag1 (Jagged1) ligand was broader, extended across both the Neurog2⁺ and *Fgf3*⁺ regions (**Figure 4—figure supplement 1J–M**). In sum, this indicates that Notch signaling pathway is involved in the specification of both the neurogenic and non-neuronal epibranchial domains.

We next assessed whether Notch signaling was altered in *Eya1*⁻/⁻ E9.5 embryos. To identify activated Notch signaling, we used an antibody that specifically recognizes the nascent N-terminus

(V1744) formed on Notch1 ICD after γ-secretase cleavage. In wildtype embryos, V1744 immunoreactivity was observed in the ectodermal and endodermal cells as well as in the neural crest (*Figure 4A and A'*), whereas the ectodermal staining was specifically lost in the *Eya1$^{-/-}$* embryos (*Figure 4B and B'*). This result may indicate that Notch1 is not proteolytically activated or that there is a reduction in the total amount of receptor. To distinguish between these two possibilities, we used an antibody (C20) that recognizes an epitope internally in the Notch1 ICD and thus monitors total amount of Notch1. The C20 immunoreactivity revealed a specific reduction of Notch1 immunoreactivity in the rostral-proximal ectodermal cells (*Figure 4D and D'*) as compared to wildtype (*Figure 4C and C'*), suggesting that the total amount of Notch1 receptor was reduced in the ectodermal cells rather than that receptor cleavage was affected. By in situ hybridization, we found that *Notch1* mRNA was expressed in the ectodermal cells at E8.5 (8ss) (*Figure 4—figure supplement 1S and T*) and E9.5 (*Figure 4—figure supplement 1U and V*), while the protein expression of Notch1 was reduced from around 8ss in the Sox2$^+$ placodal progenitors (*Figure 4—figure supplement 1O–1R'*), indicating post-transcriptional regulation of Notch1 in *Eya1$^{-/-}$* embryos. The Notch3 receptor, which was also expressed in the pharyngeal ectoderm, was not affected in the *Eya1$^{-/-}$* embryos (*Figure 4—figure supplement 1X'*). In wildtype embryos, Jag1 was expressed at high level at the PA segmental plate contact points and at a lower level in the rostral-proximal ectoderm, similar expression pattern was found in *Eya1$^{-/-}$* embryos despite the absence of PA segmental plates (*Figure 4E and F*). On the other hand, *Dll1* expression was lost in the epibranchial placode region in *Eya1$^{-/-}$* embryos (*Figure 4G and H*). The downregulation of total amount of Notch1 protein was confirmed by Western blot analysis of dissected pharyngeal regions from E9.5 *Eya1$^{-/-}$* embryos, and Eya1 protein levels were as expected non-detectable (*Figure 4M*). In keeping with reduced Notch1 receptor levels, expression of *Hey1* was significantly downregulated in the ectoderm in the *Eya1$^{-/-}$* E9.5 embryos (*Figure 4I and J*). Expression of *Hes1*, however, was much less affected (*Figure 4K and L*), which is consistent with that substantial *Hes1* expression was observed in *RBPJ$^{-/-}$* embryos (*de la Pompa et al., 1997*), suggesting that *Hes1* is also regulated by other factors. Taken together, the results indicate that *Eya1* is required for maintaining a normal level of Notch1 protein and for controlling the Notch signaling level in the ectodermal epithelium of PA2.

## Expression of Notch1 ICD rescues the non-neuronal epibranchial lineage and pharyngeal arch segmentation phenotype in *Eya1$^{-/-}$* embryos

As loss of *Eya1* resulted in reduced levels of the Notch1 receptor and lower expression of Notch downstream genes in the pharyngeal ectoderm, we next sought to explore whether restoring Notch signaling could override the phenotypic consequences of *Eya1*-deficiency. In a genetic rescue experiment, we deployed activation of Notch1 ICD in the pharyngeal ectoderm from the *Rosa26$^{N1-IC}$* mouse (*Murtaugh et al., 2003*) by *Pax2-Cre* activation (*Ohyama and Groves, 2004*). *Cre* activity and ectopic Notch1 ICD expression were revealed by EYFP and GFP staining, respectively (*Figure 5A and C*), and were both observed in the pharyngeal ectoderm (*Figure 5B and D*). A definitive first pharyngeal cleft could be found in the *Pax2-Cre; Rosa26$^{N1-IC}$; Eya1$^{-/-}$* embryos (*Figure 5F*), compared with the defective pharyngeal cleft in *Eya1$^{-/-}$*embryos (*Figure 1C*). Expression of Hey1 confirmed the upregulation of Notch signaling in the *Pax2-Cre; Rosa26$^{N1-IC}$* and *Pax2-Cre; Rosa26$^{N1-IC}$; Eya1$^{-/-}$* embryos (*Figure 5—figure supplement 1C and D*). To corroborate the restoration of pharyngeal cleft by elevating Notch signaling, we used two other Cre drivers (*B2-r4-Cre* and *Sox2-CreER*) (*Szeto et al., 2009*; *Arnold et al., 2011*) to express Notch1 ICD, and in both cases the pharyngeal segmentation phenotype was restored in *Eya1$^{-/-}$* embryos (*Figure 5—figure supplement 1G–J*). The difference between the different Cre drivers is likely attributed to slight spatial differences in the Cre activity from the different regulatory elements (*Figure 5—figure supplement 1K–V*). Moreover, expression of Notch1 ICD from the *Pax2-Cre* driver in a wildtype background, i.e. elevating Notch signaling above the normal level rather than restoring reduced Notch signaling, did not affect the normal pharyngeal cleft formation (*Figure 5E*).

To analyze the rescued phenotype in further detail, immunostaining of E-cadherin to mark the epithelium showed that in E9.5 compound mutants the pharyngeal ectoderm and endoderm made contacts and proper pharyngeal arch segmentation took place (*Figure 5J*). *Brn1* expression was also restored in the maxillary-mandibular junction of PA1 and proximal PA2 in *Pax2-Cre; Rosa26$^{N1-IC}$; Eya1$^{-/-}$* embryos (*Figure 5N and N'*). An increased number of *Sox2$^+$* progenitors could be observed

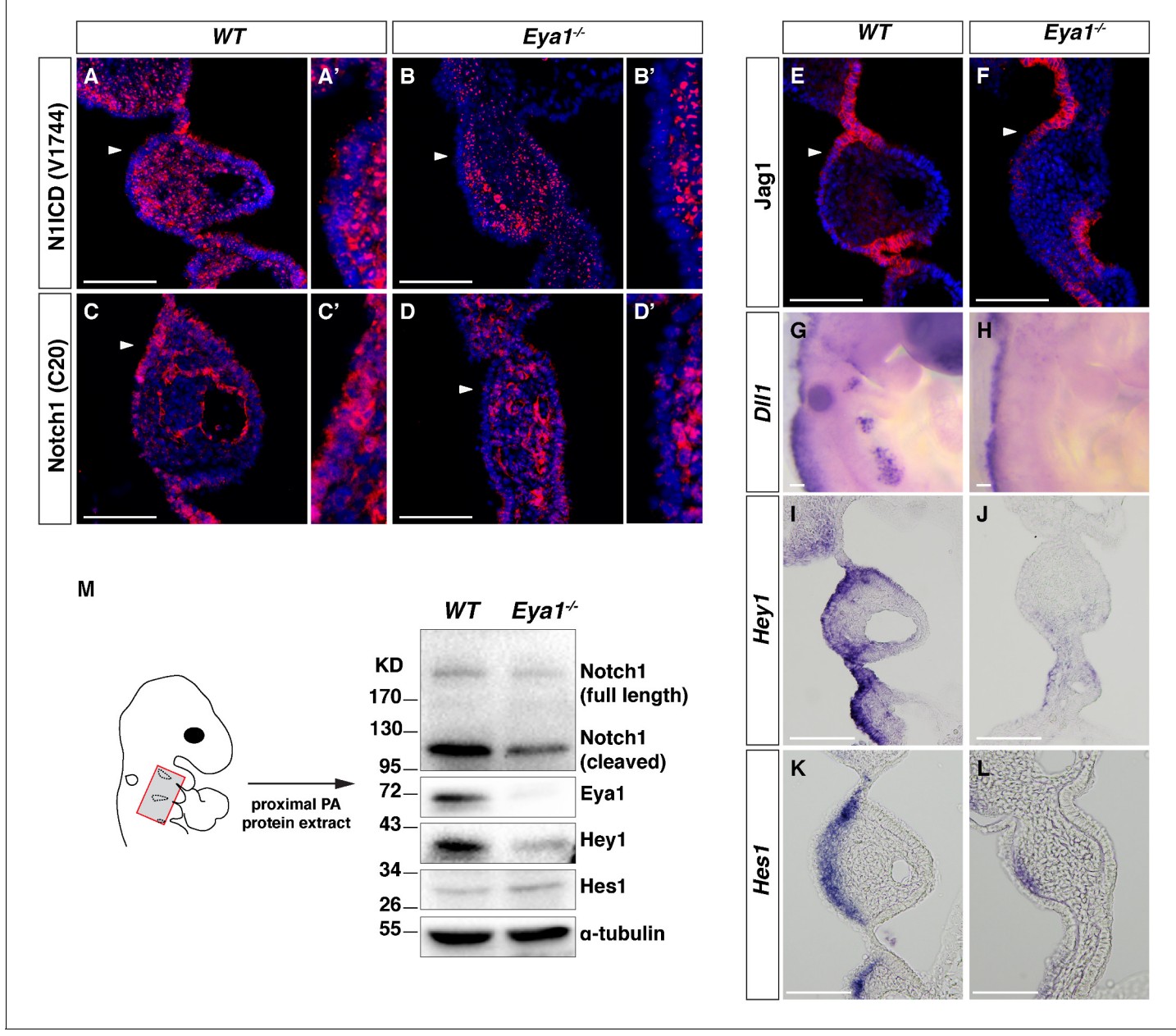

**Figure 4.** Downregulated Notch signaling in the ectoderm-derived pharyngeal epithelium of *Eya1*$^{-/-}$ embryos. (**A–D and A′–D′**) Immunostaining for Notch1 ICD by V1744, which specifically recognizes Notch1 ICD (**A–B′**); or by C20, which recognizes an internal epitope on Notch1 ICD and thus both cleaved and uncleaved Notch1 receptor (**C–D′**) on coronal sections of *WT* and *Eya1*$^{-/-}$ embryos at E9.5 (n = 4). Arrowheads indicate the rostral-proximal pharyngeal ectoderm in (**A-D**), which are magnified in (**A′-D′**). (**E–L**) Immunostaining for Jag1 (**E, F**) and in situ hybridization of *Dll1, Hey1 and Hes1* (**G–L**) on whole-mount or coronal sections of *WT* and *Eya1*$^{-/-}$ embryos at E9.5 (n ≥ 4). (**M**) Western blot analysis of tissue extracts from proximal PA2 and PA3 of *WT* and *Eya1*$^{-/-}$ embryos at E9.5 with the indicated antibodies. The red box in the embryo diagram indicates the dissected region. The Notch1 (**C20**) antibody recognizes both the full-length Notch1 receptor (Notch1 FL) and cleaved Notch1, which represents both the NEXT and ICD forms of the Notch1 receptor. One representative western blot of five. Scale bars, 100 μm.

DOI: https://doi.org/10.7554/eLife.30126.009

The following figure supplements are available for figure 4:

**Figure supplement 1.** Expression of Notch signaling factors in the ectoderm-derived pharyngeal epithelium of WT and *Eya1*$^{-/-}$ embryos.

DOI: https://doi.org/10.7554/eLife.30126.010

**Figure supplement 2.** Complete western blots for *Figure 4M*.

DOI: https://doi.org/10.7554/eLife.30126.011

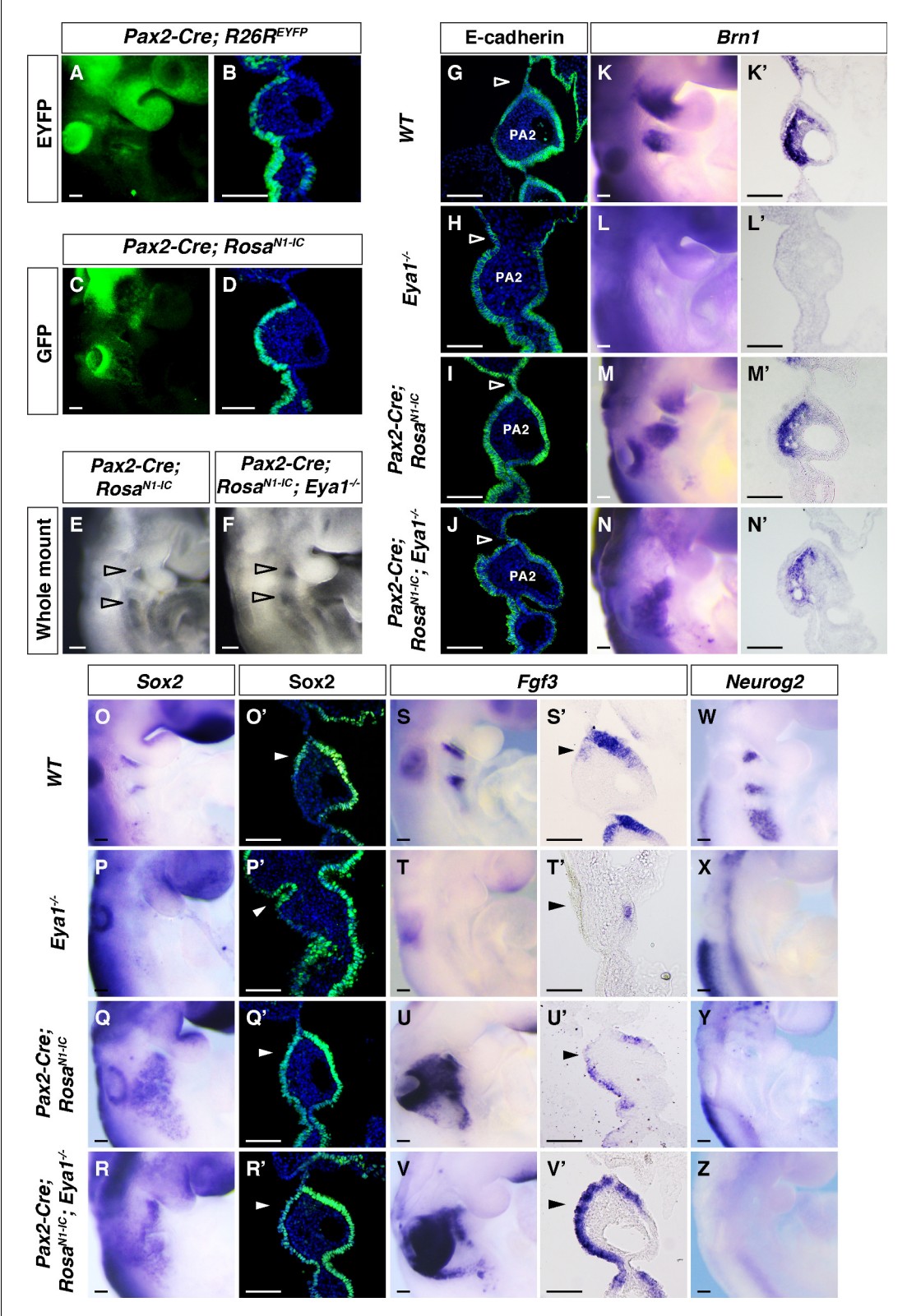

**Figure 5.** Over-expression of Notch1 ICD restores pharyngeal segmentation and Fgf3 expression in the pharyngeal epithelium of *Eya1⁻/⁻* embryos. (A–D) Whole mount fluorescence and section immunostaining for EYFP in *Pax2-Cre; R26R^EYFP* (A, B) and GFP in *Pax2-Cre;Rosa^N1-IC* (C, D) at E9.5, to visualize Cre efficiency and expression of ectopic Notch1 ICD, respectively (n = 3). (E and F) Lateral view of *Pax2-Cre; Rosa^N1-IC* and *Pax2-Cre; Rosa^N1-IC; Eya1⁻/⁻* whole-mount embryos at E9.5 (n = 5). (G–Z) Expression of for E-cadherin (G–J), *Brn1* (K–N'), Sox2 (O–R'), *Fgf3* (S–V'), and *Neurog2* (W–Z) on

*Figure 5 continued on next page*

*Figure 5 continued*

whole-mount or coronal sections of *WT*, *Eya1^-/-^*, *Pax2-Cre; Rosa^N1-IC^* and *Pax2-Cre; Rosa^N1-IC^; Eya1^-/-^* embryos at E9.5 (n ≥ 3). Open arrowheads indicate positions of the pharyngeal clefts. Arrowheads indicate the rostral-proximal pharyngeal ectodermal cells. Scale bars, 100 μm.

DOI: https://doi.org/10.7554/eLife.30126.012

The following figure supplement is available for figure 5:

**Figure supplement 1.** Over-expression of Notch1 ICD by Pax2-Cre, *B2-r4-Cre* and *Sox2-CreER* restored pharyngeal segmentation and morphogenesis in *Eya1^-/-^* embryos.

DOI: https://doi.org/10.7554/eLife.30126.013

in the entire ectodermal epithelium of PA2 of the *Pax2-Cre; Rosa26^N1-IC^* and *Pax2-Cre; Rosa26^N1-IC^; Eya1^-/-^* (*Figure 5R and R'*). Furthermore, Pax2-*Cre* mediated Notch1 ICD expression rescued the expression of *Fgf3* at E9.5 (*Figure 5V and V'*). The reduced expression of *Spry*1 in the Eya1^-/-^ embryos (*Figure 2Q*) was restored upon Notch1 ICD expression (*Figure 5—figure supplement 1E and F*), further supporting that Fgf signaling is regulated by Notch. Importantly, however, Neurog2 expression was inhibited by Notch1 ICD expression (*Figure 5Y and Z*), which is in keeping with a role for high Notch signaling in blocking neuronal differentiation in cranial placodes (*Lassiter et al., 2014*). Collectively, the results indicate that induced Notch signaling can restore Fgf downstream signaling, rescue the morphological and functional defects of pharyngeal epithelium caused by loss of Eya1 and that Notch signaling is sufficient to promote the formation of the non-neurogenic epibranchial cell population.

## Eya1 dephosphorylates Notch1 ICD leading to enhanced Notch1 ICD stability

To address the link between Eya1 and the level of Notch1 receptor, we first assessed the interactions between Eya1 and Notch1 ICD proteins in transfected 293T cells. Co-immunoprecipitation (Co-IP) experiments revealed an interaction between Flag-Eya1 and Myc-Notch1 ICD (*Figure 6A*), and the C-terminal domain of Eya1 showed a strong interaction with Myc-Notch1 ICD while the N-terminal domain exhibited a very weak or no interaction (*Figure 6B*). Furthermore, co-transfected cells with a constant amount of Myc-Notch1 ICD and increasing amounts of Flag-Eya1 showed that the amount of Notch1 ICD was elevated proportionally to the increase in the amount of Eya1 (*Figure 6C*), suggesting that Eya1 enhances the stability of Notch1 ICD. We also observed increased amounts of full length Notch1 receptor and a membrane-tethered form mimicking an ADAM-processed Notch1 receptor (Notch1ΔE) in response to Eya1 in cells that were treated with γ-secretase inhibitor (DAPT) to block S3-mediated cleavage of the receptor (*Figure 6D*). To assess the half-life of Notch1 ICD, we blocked protein synthesis by cycloheximide, and the short half-life of Notch1 ICD (approximately 5 hr), which is in keeping with previous reports (*Gustafsson et al., 2005*), was significantly extended (more than 10 hr) when Eya1 was co-transfected (*Figure 6E and F*).

Eya1 contains both a tyrosine and a threonine phosphatase activity that co-localize in the C-terminal domain (*Li et al., 2017*), and to explore which activity was linked to Notch1, we analyzed the effects of Eya1 mutations that abolish phosphatase activity. In contrast to the prolongation of Notch1 ICD half-life by wildtype Eya1, two Eya1 mutants DYY and Y4, which largely affect the threonine phosphatase activity (*Okabe et al., 2009*) did not stabilize Notch1 ICD (*Figure 6E and F*) and this was the case also for Eya1D327N, which affects both the tyrosine and threonine phosphatase activities (*Li et al., 2017*). To corroborate the notion that the Eya1 threonine phosphatase activity was involved, we explored whether the amount of threonine-, serine- or tyrosine-phosphorylation of Notch1 ICD was altered by Eya1. Analysis using anti-phosphopeptide-specific antibodies for threonine, serine and tyrosine revealed that in the presence of Eya1, the total amount of Notch1 ICD was increased but the amount of threonine-phosphorylated Notch1 ICD was reduced (*Figure 6G*). In contrast, the serine-phosphorylated Notch1 ICD was not affected (*Figure 6—figure supplement 1B*), and tyrosine-phosphorylated Notch ICD was hardly detected (*Figure 6—figure supplement 1C*). In keeping with these data, in vitro phosphatase experiments, using purified Flag-Eya1 and Flag-Notch1 ICD (*Figure 6—figure supplement 1A*), showed that in the presence of Eya1 the amount of threonine-phosphorylated Notch1 ICD was reduced (*Figure 6H*). As the stability of Notch1 ICD is regulated by phosphorylation and ubiquitination, in particular by the E3 ubiquitin

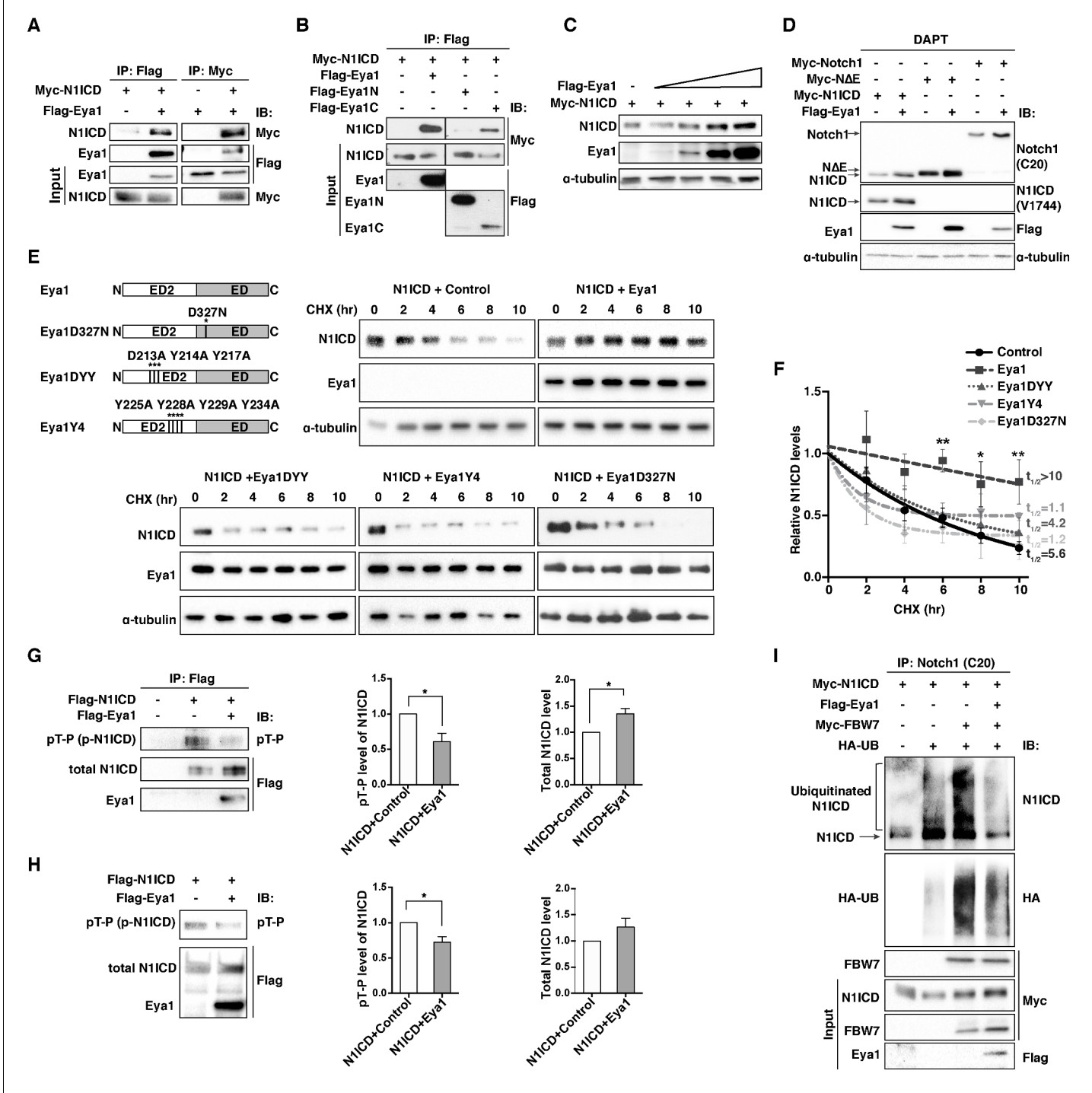

**Figure 6.** Eya1 stabilizes and dephosphorylates Notch1 ICD. (A) Co-immunoprecipitation (Co-IP) analysis using 293T cells with (+) or without (-) transfected *Myc-Notch1 ICD* and *Flag-Eya1*, as indicated (n = 6). Pull-down assays (IP) using anti-Flag or anti-Myc antibodies. Western blot analyses (IB) using anti-Myc or anti-Flag antibodies. Input was 5% of the amount of proteins used for IP. (B) IP analysis using 293T cells transfected with *Myc-Notch1 ICD*, full-length *Eya1*, the N- or C-terminal portions of *Eya1* (Eya1N and Eya1C, respectively) (n = 3). Input was 5% of the amount of proteins used for IP. (C) Increased amount of *Flag-Eya1* was transfected into 293T cells with control or *Myc-Notch1 ICD*, leading to increased level of Notch1 ICD at higher amounts of Eya1. α-tubulin was used as internal control (n = 3). (D) Western blot analysis of 293T cells transfected with different plasmids as indicated. Cells were treated with 10 μM γ-secretase inhibitor (DAPT) for 48 hr prior to harvesting (n = 4). (E) Western blot analysis of cycloheximide (CHX) treated cells transfected with *Myc-Notch1 ICD* and wild-type or mutant *Eya1* cDNAs at indicated time points using anti-Notch1 ICD (V1744), anti-Flag (Eya1), or anti-α-tubulin antibodies. Eya1 has a conserved transcriptional activation domain ED and an N-terminal ED2 domain. The DYY and Y4 mutants

*Figure 6 continued on next page*

*Figure 6 continued*

abolished the threonine phosphatase activity, while the D327N mutant abolished both the tyrosine- and threonine-phosphatase functions of Eya1 (n ≥ 3). (F) Summary graph showing quantification of the average Notch1 ICD protein levels relative to α-tubulin in experiments shown in (E). The relative protein levels were normalized to time point zero, fitted with one-phase decay, and the half-life ($t_{1/2}$) calculated. Error bars represent SEM. *p<0.05. Statistical significance between two groups was determined by unpaired two-tailed Student's t-test. (G) Western blot analysis of phosphorylation status of Notch1 ICD. Flag-Notch1 ICD was immunoprecipitated with anti-FLAG M2 antibody with (+) or without (-) transfected *Flag-Eya1*, using pT-P antibodies (anti-phospho-Threonine-Proline antibody) or anti-Flag targeting the total Notch1 ICD. Graphs show quantification of the average levels of phosphorylated or total Notch1 ICD in the presence of control or Eya1 (n = 5). (H) Analysis of dephosphorylation of Notch1 ICD by Eya1 by an in vitro phosphatase assay. Flag-Notch1 ICD and Flag-Eya1 purified from 293T cells were incubated in vitro, after which the phosphorylation status of Notch1 ICD was examined by immunoblotting with anti-pT-P and anti-Flag antibodies. Graphs show quantification of the average levels of phosphorylated or total Notch1 ICD incubated with or without Eya1 (n = 4). (G and H) Error bars represent SEM, and p values were calculated using one sample t-test, *p<0.05. (H) Ubiquitination of Notch1 ICD was reduced in the presence of Eya1. Lysates from 293T cells transfected with indicated plasmids were treated with 20 μM MG132 for 6 hr before lysis (n = 3). The lysate were immunoprecipitated with anti-Notch1 (C20) and immunoblotted with anti-Notch1 ICD (V1744), anti-HA, anti-Myc and anti-Flag. Input was 5% of the amount of proteins used for IP.
DOI: https://doi.org/10.7554/eLife.30126.014

The following source data and figure supplements are available for figure 6:

**Source data 1.** Source data relating to *Figure 6E*.
DOI: https://doi.org/10.7554/eLife.30126.017
**Source data 2.** Source data relating to *Figure 6F*.
DOI: https://doi.org/10.7554/eLife.30126.018
**Source data 3.** Source data relating to *Figure 6G*.
DOI: https://doi.org/10.7554/eLife.30126.019
**Figure supplement 1.** Eya1 does not affect serine or tyrosine phosphorylation level of Notch1 ICD.
DOI: https://doi.org/10.7554/eLife.30126.015
**Figure supplement 2.** Complete western blots for *Figure 6* and *Figure 6—figure supplement 1*.
DOI: https://doi.org/10.7554/eLife.30126.016

ligase Fbw7 (*Oberg et al., 2001*), we examined the impact of Eya1 on the turnover of Notch1 ICD. By immunoprecipitation experiments, we demonstrated that the ubiquitinated-Notch1 ICD level was enhanced by Fbw7, but reduced significantly in the presence of Eya1 (*Figure 6I*). The level of ubiquitination, assessed by anti-HA antibody, was also reduced by Eya1 (*Figure 6I*). In conclusion, the data show that Eya1 interacts with Notch1 ICD via its C-terminal domain and that Eya1 acts as a threonine phosphatase on Notch1 ICD, controlling its stability.

To identify the target dephosphorylated threonine residue on Notch1 ICD, we noted that there was a sequence similarity between the region around T2122 in Notch1 ICD and the region neighboring T58 in c-Myc (*Wei et al., 2005*) (*Figure 7A*), an established target threonine residue for the Eya1 threonine phosphatase activity (*Xu et al., 2014*; *Li et al., 2017*). To test the importance of T2122 in relation to Eya1 phosphatase activity and Notch1 ICD stability, we generated phospho-mimetic (T2122D) and phospho-dead (T2122A) versions of Notch1 ICD, i.e. mutant versions that mimic the phosphorylated form and which cannot be phosphorylated, respectively (*Figure 7A*). The phospho-dead Notch1 ICD[T2122A] mutant could interact with Eya1 (*Figure 7—figure supplement 1A*), and its threonine phosphorylation status was not altered by Eya1 (*Figure 7B*), indicating that T2122 is the phosphorylation site targeted by Eya1. In line with this, the amount and threonine phosphorylation of in vitro phosphorylated Notch1 ICD[T2122A] was largely unaffected by Eya1 (*Figure 7C*). The only other candidate site for Eya1 phosphatase activity in Notch1 ICD (T2487) (*Figure 7—figure supplement 1B*), which is also a target of Fbw7 (*O'Neil et al., 2007*), was not affected by Eya1 (*Figure 7B* and *Figure 7—figure supplement 1C*) and therefore unlikely to be an in vivo target. Finally, we assessed the performance of the phospho-mimetic and phospho-dead Notch1 ICD mutants with regard to stability in response to Eya1. The results from the cycloheximide treatment experiments showed that the phospho-dead Notch1 ICD[T2122A] was considerably more stable (half-life = 10 hr) than control Notch1 ICD, whereas Eya1 did not significantly alter the stability of Notch1 ICD[T2122A] (*Figure 7D*). In contrast, the phospho-mimetic Notch1 ICD[T2122D] mutant displayed a short half-life (5 hr), at par with control Notch1 ICD, and its half-life was not affected by Eya1 (*Figure 7E*). In sum, these results show that threonine 2122 plays a crucial role in regulating Notch1 ICD stability and is the likely target amino acid residue for the threonine phosphatase activity of Eya1.

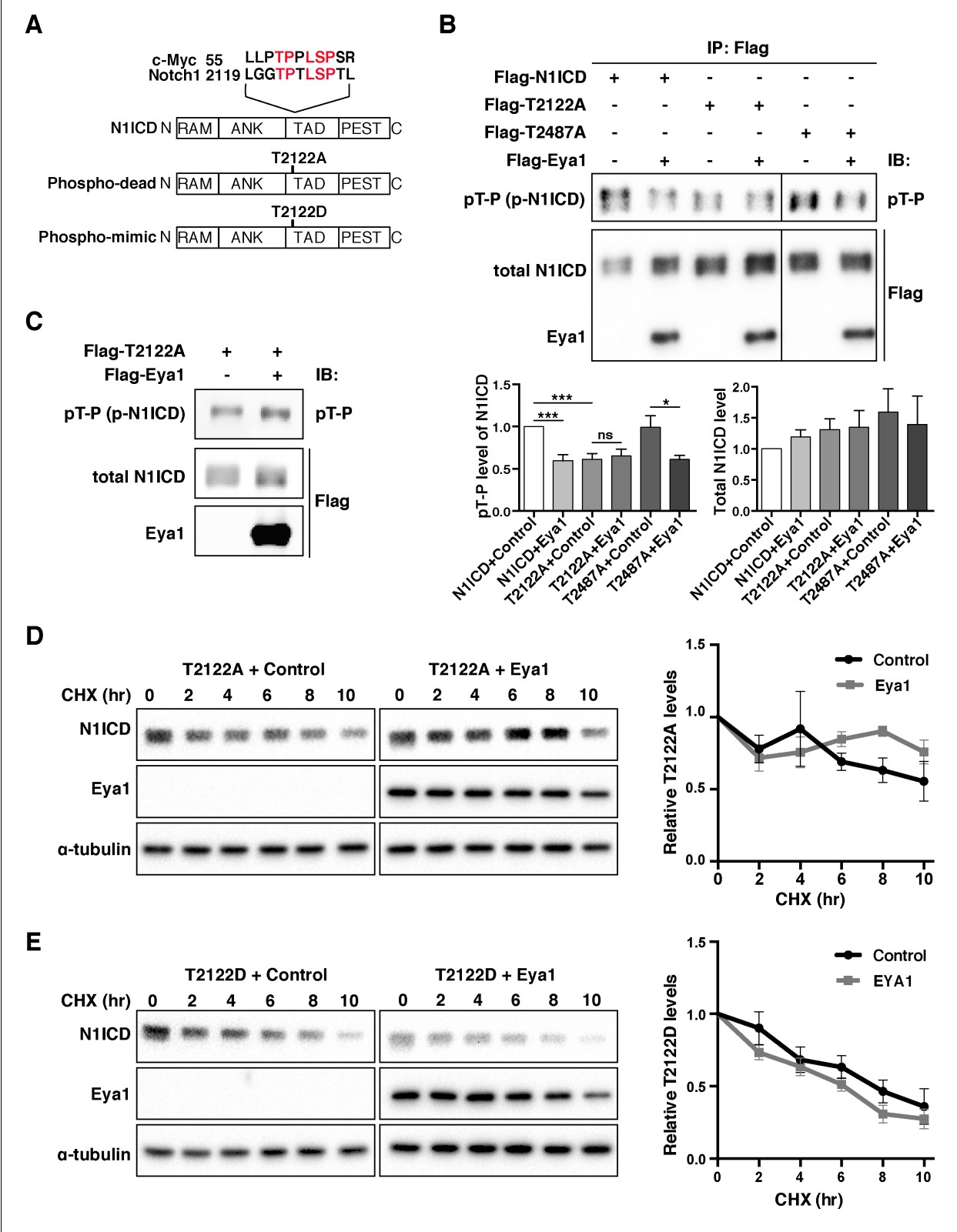

**Figure 7.** Eya1 targets the T2122 phosphorylation site of Notch1 ICD. (**A**) Schematic diagram showing Notch1 ICD protein domains. Alignment of Notch1 ICD peptide sequence (from a.a. 2119) with c-Myc (from a.a. 55), positions of the phospho-dead (T2122A) and phospho-mimic (T2122D) mutations are indicated. Abbreviations: RAM, Rbp-associated molecule domain; ANK, ankyrin repeat domain; TAD, transcription activation domain; PEST, domain rich in proline, glutamic acid, serine and threonine. (**B**) Phosphorylation analysis of WT Notch1 ICD, Notch1 ICD^T2122A, and Notch1

*Figure 7 continued on next page*

*Figure 7 continued*

ICD$^{T2487A}$ by Eya1 phosphatase (n $\geq$ 3). Notch1 ICD was immunoprecipitated with anti-FLAG M2 antibody and analyzed by immunoblotting with anti-pT-P and anti-Flag antibodies. Graphs show quantification of the average levels of phosphorylated or total Notch1 ICD, Notch1 ICD$^{T2122A}$ and Notch1 ICD$^{T2487A}$ in the presence of control or Eya1. Statistical significance between two groups was determined by one sample t-test. Error bars represent SEM. ***p<0.001; *p<0.05; ns, not significant. (C) In vitro phosphatase treatment of Notch1 ICD$^{T2122A}$ by Eya1. Flag-T2122A and Flag-Eya1 purified from 293T cells were incubated in vitro, and then examined by immunoblotting with anti-pT-P and anti-Flag antibodies (n = 3). (D and E) Mutant Notch1 ICD protein stability analysis (n = 3). 293T cells transfected with indicated plasmids were treated with CHX. At indicated time points, cells were lysed and examined by immunoblotting with anti-Notch1 ICD (V1744), anti-Flag, or anti-$\alpha$-tubulin antibodies. Graphs show quantification of the average Notch1 ICD$^{T2122A}$ or Notch1 ICD$^{T2122D}$ expression levels relative to $\alpha$-tubulin. The relative protein levels were normalized to time point zero. No significance was observed between the groups transfected with control or Eya1 using unpaired two-tail t-test. Error bars represent SEM.
DOI: https://doi.org/10.7554/eLife.30126.020

The following source data and figure supplements are available for figure 7:

**Source data 1.** Source data relating to *Figure 7B*.
DOI: https://doi.org/10.7554/eLife.30126.023
**Source data 2.** Source data relating to *Figure 7D*.
DOI: https://doi.org/10.7554/eLife.30126.024
**Source data 3.** Source data relating to *Figure 7E*.
DOI: https://doi.org/10.7554/eLife.30126.025
**Figure supplement 1.** Eya1 does not affect the T2487 phosphorylation site of Notch1 ICD, related to *Figure 7*.
DOI: https://doi.org/10.7554/eLife.30126.021
**Figure supplement 2.** Complete western blots for *Figure 7* and *Figure 7—figure supplement 1*.
DOI: https://doi.org/10.7554/eLife.30126.022

## Discussion

In this study, we have uncovered the presence of groups of non-neuronal placodal cells among the epibranchial placodes. We show that a common Sox2$^+$ cell population produces both delaminating neurons and a novel non-neuronal placodal cell population important for PA development and architecture. To investigate the molecular basis of epibranchial placode cell fate decisions, we have identified Notch1 ICD as a novel substrate for Eya1 phosphatase. The choice whether to become a neuronal or non-neuronal cell is regulated by Eya1 and Notch, where Eya1 acts upstream of Notch via its threonine phosphatase function (summarized in *Figure 8*).

### The discovery of a non-neuronal epibranchial cell population reveals a bipotential differentiation program in epibranchial placodes

The epibranchial placodes delaminate cells that migrate and undergo neuronal differentiation, providing sensory neurons for cranial nerves VII, IX and X. The epibranchial placodes, along with the trigeminal placode, are called neurogenic placodes and are distinct from other placodes, such as the olfactory and otic placodes, which generate both neurons and non-neuronal supporting cells (*Beites et al., 2005*; *Patthey et al., 2014*). Previous findings in chick reveal that *Sox3* not only marks the classically-defined epibranchial placodes but also a larger domain that ceases to produce neurons (*Ishii et al., 2001*; *Abu-Elmagd et al., 2001*; *Tripathi et al., 2009*). In this report, we provide evidence that the epibranchial placodes generate two lineage-related cell types from a common Sox2-expressing progenitor cell pool present at around E8.0. These cells undergo a stepwise differentiation and segregation into two cell types: the previously characterized Neurog2$^+$ delaminating neurons (*Fode et al., 1998*) as well as Sox2$^+$ and Fgf3$^+$ non-neuronal cells. The segregation to the neuronal and non-neuronal fates appears to be gradual. At E8.5, the placodal progenitors have already initiated the expression of *Sox2, Fgf3*, and *Neurog2*. Half a day later, the Sox2 single positive, Neurog2 single positive, and Sox2-Neurog2 double positive placodal progenitors are distributed in a salt and pepper pattern, implying that these cells may undergo a specification and differentiation process but are still intermingled. The Neurog2$^+$ population goes on to form the delaminating neurons, also taking on Islet1 expression and clusters separately from the non-neuronal progenitors (summarized in *Figure 3U*). This new bipotential model for epibranchial placode differentiation makes the epibranchial placode more similar to other cranial placodes, which seed both neuronal and supporting non-neuronal cells, such as the olfactory placodes. The findings also have an evolutionary bearing, and not only reveal a similarity to the chick epibranchial placode as

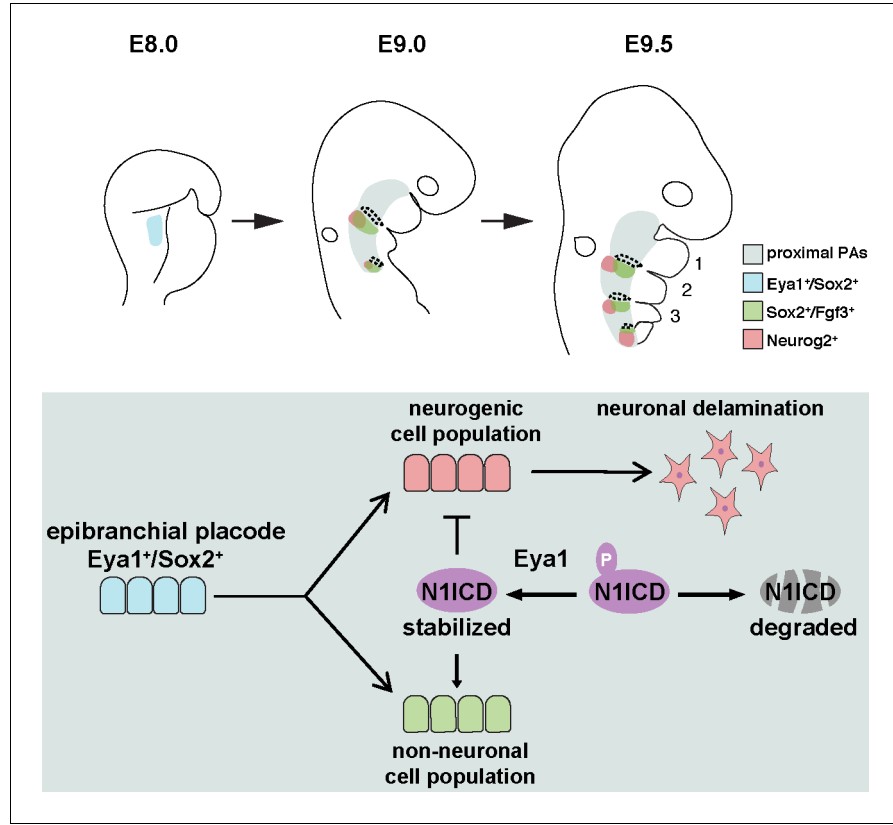

**Figure 8.** Model illustrating the role of Eya1 and Notch in regulating neurogenic and non-neuronal lineages during epibranchial placode development. During PA development, the Eya1+/Sox2+ placodal progenitors formed at E8.0 are separated into two distinct cell populations by E9.0: the Neurog2+ neurogenic cells and the Sox2+/Fgf3+ non-neuronal cells in the proximal PA region, and with further spatial separation at E9.5. Our findings indicate that Notch1 ICD is dephosphorylated and stabilized by Eya1. The Notch1 ICD level is critical to control the cell fate decision for these two distinct cell populations and to regulate neurogenesis, pharyngeal segmentation and development of proximal arches.
DOI: https://doi.org/10.7554/eLife.30126.026

discussed above, but also to a novel placode located adjacent to the geniculate placode that was recently discovered in the chick (*O'Neill et al., 2012*). This placode is also derived from Sox2+ cells and gives rise to the mechanosensory paratympanic organ (PTO) in the chick middle ear (*O'Neill et al., 2012*). The PTO will also delaminate neuronal precursors, indicating that in the chick, a placode with a dual fate exists. The PTO is not unique to the chick but has been found in several species of birds (*Giannessi et al., 2013*) and some reptiles also harbor PTO-like structures (*Neeser and von Bartheld, 2002*), but these have not been reported in mammals (*Giannessi et al., 2013*).

## The non-neuronal epibranchial placodal cell population regulates pharyngeal arch development

It may be asked what the function of the non-neuronal epibranchial placodal cell population is. Differentiation of the non-neuronal epibranchial placodal cells is spared in the *Six1-/-* embryos, whereas in the *Eya1-/-* embryos, both neuronal differentiation as well as Fgf expression in the non-neuronal epibranchial placode are affected. Therefore, one may assume that phenotypes observed only in *Eya1-/-* embryos can be attributed to consequences of derailed differentiation in the non-neuronal lineage. It appears reasonable to assume that dysregulation of Fgf signaling downstream of Eya1 and Notch plays a role for the aberrant pharyngeal arch development, as mutations in *Fgfr1, Fgf8* and *Spry1* affect these processes (*Trokovic et al., 2003*; *Trokovic et al., 2005*; *Trumpp et al.,*

*1999*; *Tucker et al., 1999*; *Abu-Issa et al., 2002*; *Simrick et al., 2011*), but more work is required to precisely identify which Fgf ligand is most important. Similarly, it may be reasoned that the location of the non-neuronal placodal cells precisely where segmentation of the PA occurs would make them suitable candidates as signaling centers for this process, but additional research is required to stringently test their role in the formation of proximal PA. In addition to the geniculate placode, the petrosal and nodose placodes also consist of Sox2$^+$ and Fgf3$^+$ non-neuronal placodal cells, as well as Neurog2$^+$ neurogenic cells (*Figure 3—figure supplement 1F and G*).

## An Eya1-Notch axis in epibranchial placode differentiation

We provide several lines of evidence that Notch acts downstream of Eya1 in craniofacial development. Firstly, Eya1 positively regulates Notch signaling activity during pharyngeal epithelium development, and in the absence of *Eya1*, the level of Notch1 receptor as well as expression of the Notch downstream gene *Hey1* was down-regulated in the pharyngeal epithelium. Secondly, ectopic expression of Notch1 ICD in Pax2-expressing cells was sufficient to rescue the PA phenotypes caused by loss of Eya1. Importantly, the phenotypic rescue by Notch1 ICD expression included restored expression of *Fgf3* and *Spry1* in the epibranchial surface, indicating that Notch1 ICD is sufficient to activate the expression of *Fgf3* and *Spry1* without *Eya1* in the pharyngeal ectoderm. Interestingly, expressing supra-physiological levels of Notch1 ICD, as determined by the elevated levels of *Hey1* (*Figure 5—figure supplement 1C and D*) expression, did not generate a phenotype, which is in contrast to other situations of hyperactivated Notch signaling, for example in acute lymphoblastic T-cell leukemia (*Weng et al., 2004*).

Our analysis identifies specific roles for Notch signaling in the different steps of epibranchial placode differentiation. We confirm previous reports that Notch signaling exerts a negative effect on neuronal fate specification, differentiation and delamination from the epibranchial placode (*Lassiter et al., 2014*) as neuronal differentiation, as judged by Neurog2 and Islet1 expression, was largely ablated upon Notch1 ICD overexpression. In the neuronal differentiation program, which also requires Eya1, Notch signaling needs to be turned off in order for neurons to differentiate. As concerns the non-neuronal epibranchial cells, we define a new role for Notch in promoting the differentiation of the Sox2$^+$/ Fgf3$^+$ cells (summarized in *Figure 8*). Here, Notch signaling needs to be sustained by the Eya1 threonine phosphatase activity. It will be interesting to explore whether the Eya1-Notch axis may be important also in other organs, where both Eya1 and Notch are expressed, and in support of such a notion, it has been observed that in the developing lung epithelium of *Eya1$^{-/-}$* embryos, the expression of Notch1 ICD, *Hes1*, and *Hes5* was reduced (*El-Hashash et al., 2011*).

## Eya1 positively regulates Notch signaling via dephosphorylation of Notch1 ICD

We provide evidence that Eya1 acts as threonine phosphatase for Notch1 ICD, and while a number of kinases responsible for phosphorylating Notch ICDs have been identified, including CDK8, GSK3β, NEMO, aPKC and PIM kinase (*Foltz et al., 2002*; *Fryer et al., 2004*; *Ishitani et al., 2010*; *Sjöqvist et al., 2014*; *Santio et al., 2016*), this is to the best of our knowledge the first demonstration of a phosphatase acting on Notch ICD. Notch1 ICD interacts with the C-terminal domain of Eya1, which harbors the threonine kinase activity (*Li et al., 2017*). The target site for the Eya1 threonine phosphatase activity is likely to be T2122 in Notch1 ICD, as the phospho-dead Notch1 ICD$^{T2122A}$ mutant was not dephosphorylated by Eya1. Furthermore, the sequence composition around T2122 in Notch is strikingly similar to a domain in Myc, which harbors a threonine residue (T58) that is also dephosphorylated by Eya1 (*Xu et al., 2014*; *Li et al., 2017*). The notion that dephosphorylation by Eya1 positively affects Notch1 ICD stability is supported by several lines of evidence. Notch1 ICD half-life was considerably enhanced by the presence of Eya1 and conversely, *Eya1$^{-/-}$* embryos showed reduced levels of the Notch1 receptor, accompanied by reduced expression of the Notch downstream gene *Hey1*. Eya1 is likely to exert an effect also on the full length Notch1 receptor, as the amount of Notch1 receptor was elevated after Eya1 stimulation under conditions when γ-secretase-mediated cleavage was blocked. Mutations in Eya1 that affect the threonine phosphatase activity in the C-terminal region or substrate-interacting domains in the N-terminal region failed to prolong the half-life of Notch1 ICD. Furthermore, the phospho-dead version of Notch1 ICD (Notch1 ICD$^{T2122A}$) showed enhanced stability and was not affected by Eya1, whereas a phospho-

mimetic mutant (Notch1 ICD$^{T2122D}$) was short-lived, both in the presence and absence of Eya1 (*Figure 7*). Collectively, these data provide a strong correlation between the phosphorylation status at T2122 and Notch1 ICD stability.

The identification of Eya1 as a phosphatase for T2122 raises the question of the nature of the kinase phosphorylating this site. The sequences around T2122 in Notch1 ICD are quite similar to sequences in Myc, cyclin E and c-Jun (*Wei et al., 2005*) (*Figure 7—figure supplement 1B*), and the threonine residues at the positions corresponding to T2122 in Myc and c-Jun can be phosphorylated by GSK3β (*Welcker et al., 2004*; *Wei et al., 2005*). This makes GSK3β a strong candidate as a kinase for T2122 in Notch1 ICD and GSK3β has indeed been shown to phosphorylate Notch ICDs (*Foltz et al., 2002*). The conserved domain around T2122 also constitutes a binding site for the Fbw7 E3 ubiquitin ligase in c-Jun where phosphorylation of the corresponding threonine residue in the c-Jun phosphodegron motif stimulates ubiquitination and proteasome-mediated degradation (*Wei et al., 2005*). We therefore propose that dephosphorylation of T2122 by Eya1 leads to decreased ubiquitination and therefore enhanced stability of Notch1 ICD.

In conclusion, our data shed new light onto the process of PA and epibranchial placode differentiation by providing evidence for a non-neuronal developmental trajectory from the epibranchial placodes. The non-neuronal epibranchial placodal cells are important for PA and cleft formation and link the placodal and PA developmental processes. The data also reveal that this more complex differentiation program in the epibranchial placodes is regulated by a novel Eya1-Notch axis and Notch1 ICD phosphorylation status.

# Materials and methods

## Key resources table

| Reagent type (species) or resource | Designation | Source or reference | Identifiers | Additional information |
|---|---|---|---|---|
| antibody | Goat polyclonal anti-Sox2 | Neuromics (Minnesota, USA) | GT15098-100, RRID: AB_2195800 | 1/500, IHC |
| antibody | Goat polyclonal anti-Sox3 | R and D systems (Minnesota, USA) | AF2569, RRID: AB_2239933 | 1/500, IHC |
| antibody | Rat monoclonal anti-Jagged1 | DSHB (Iowa, USA) | Ts1.15h, RRID: AB_528317 | 1/300, IHC |
| antibody | Rabbit monoclonal anti-E-cadherin (24E10) | Cell Signaling Technology (Massachusetts, USA) | 3195, RRID: AB_2291471 | 1/1000, IHC |
| antibody | Mouse polyclonal anti-Eya1(A01) | Abnova (Taiwan) | H00002138-A01, RRID: AB_563241 | 1/300, IHC, WB |
| antibody | Goat polyclonal anti-Notch1 (C20) | Santa Cruz Biotechnology (Texas, USA) | sc-6014; RRID: AB_650336 | 1/400, IHC, WB, IP |
| antibody | Rabbit polyclonal anti-Cleaved Notch1 | Abcam (Hong Kong) | ab52301, RRID: AB_881726 | 1/500, IHC, WB |
| antibody | Rabbit polyclonal anti-Cleaved Notch1 (Val1744) | Cell Signaling Technology | 2421S, RRID: AB_2314204 | 1/500, IHC, WB |
| antibody | Mouse monoclonal anti-Notch3 | BioLegend (California, USA) | 130502, RRID: AB_1227735 | 1/1000, IHC |
| antibody | Mouse monoclonal anti-Phospho-Threonine-Proline | Cell Signaling Technology | 9391S, RRID: AB_331801 | 1/3000, WB |
| antibody | Rabbit monoclonal anti-Phospho-Serine-Proline-Proline motif [pSPP] | Cell Signaling Technology | Cat#14390 | 1/1000, WB |
| antibody | Mouse monoclonal anti-p-Tyrosine (PY99) | Santa Cruz Biotechnology | sc-7020, RRID:AB_628123 | 1/1000, WB |
| antibody | Mouse monoclonal anti-Neurog2 | R and D systems | MAB3314, RRID:AB_2149520 | 1/1000, IHC |

*Continued on next page*

*Continued*

| Reagent type (species) or resource | Designation | Source or reference | Identifiers | Additional information |
|---|---|---|---|---|
| antibody | Mouse polyclonal anti-Islet1 | DSHB | PCRP-ISL1-1A9, RRID:AB_2618775 | 1/400, IHC |
| antibody | Rabbit polyclonal anti-GFP | Abcam | ab6556, RRID:AB_305564 | 1/1000, IHC |
| antibody | Rabbit polyclonal anti-Hes1 | Abcam | ab71559, RRID:AB_1209570 | 1/500, WB |
| antibody | Rabbit polyclonal anti-Hey1 | Abcam | AB22614, RRID:AB_447195 | 1/500, WB |
| antibody | Mouse monoclonal anti-GAPDH [6C5] | Abcam | Ab8245, RRID:AB_2107448 | 1/10000, WB |
| antibody | Mouse monoclonal anti-FLAG(R) M2 | Sigma-Aldrich (Missouri, USA) | F1804, RRID:AB_262044 | 1/500 for IP, 1/4000 for WB |
| antibody | Rabbit monoclonal anti-Flag | Sigma-Aldrich (Missouri, USA) | F7425, RRID:AB_439687 | 1/4000, WB |
| antibody | Mouse monoclonal anti-c-Myc (9E10) | Santa Cruz Biotechnology | sc-40, RRID:AB_627268 | 1/1000, WB |
| antibody | Mouse monoclonal anti-Tubulin | DSHB | Cat#AA4.3-s; RRID: AB_579793 | 1/5000, WB |
| antibody | Rabbit polyclonal anti-HA-probe | Santa Cruz Biotechnology | sc-805, RRID:AB_631618 | 1/1000, WB |
| antibody | Donkey anti-Goat IgG (H + L) Cross-Adsorbed Secondary Antibody, Alexa Fluor 488 | Thermo Fisher Scientific (Hong Kong) | A-11055, RRID:AB_2534102 | 1/500, IHC |
| antibody | Donkey anti-Rabbit IgG (H + L) Highly Cross-Adsorbed Secondary Antibody, Alexa Fluor 488 | Thermo Fisher Scientific | A21206, RRID:AB_2535792 | 1/500, IHC |
| antibody | Donkey anti-Rat IgG (H + L) Highly Cross-Adsorbed Secondary Antibody, Alexa Fluor 488 | Thermo Fisher Scientific | A-21208, RRID:AB_2535794 | 1/500, IHC |
| antibody | Donkey anti-Mouse IgG (H + L) Highly Cross-Adsorbed Secondary Antibody, Alexa Fluor 555 | Thermo Fisher Scientific | A-31570, RRID:AB_2536180 | 1/500, IHC |
| antibody | Donkey anti-Goat IgG (H + L) Cross-Adsorbed Secondary Antibody, Alexa Fluor 594 | Thermo Fisher Scientific | A-11058, RRID:AB_2534105 | 1/500, IHC |
| antibody | Donkey anti-Mouse IgG (H + L) Highly Cross-Adsorbed Secondary Antibody, HRP | Thermo Fisher Scientific | A16017, RRID:AB_2534691 | 1/5000, WB |
| antibody | Donkey anti-Rabbit IgG (H + L) Highly Cross-Adsorbed Secondary Antibody, HRP | Thermo Fisher Scientific | A16035, RRID:AB_2534709 | 1/5000, WB |
| antibody | Donkey anti-goat HRP Conjugate | Santa Cruz Biotechnology | sc-2020, RRID:AB_631728 | 1/5000, WB |
| antibody | Anti-Digoxigenin-AP, Fab fragments | Roche (Germany) | Cat#11093274910 | 1/2000, ISH |
| peptide, recombinant protein | 3XFLAG peptide | Sigma-Aldrich (Missouri, USA) | Cat# F4799 | |
| chemical compound, drug | X-tremeGENE 9 DNA Transfection Reagent | Roche | Cat#6365779001 | |
| chemical compound, drug | Proteinase K | Sigma-Aldrich (Missouri, USA) | Cat#P6556 | |

*Continued on next page*

*Continued*

| Reagent type (species) or resource | Designation | Source or reference | Identifiers | Additional information |
|---|---|---|---|---|
| chemical compound, drug | Cycloheximide | Sigma-Aldrich (Missouri, USA) | Cat#C7698 | |
| chemical compound, drug | Tamoxifen | Sigma-Aldrich (Wisconsin, USA) | Cat#06734 | |
| chemical compound, drug | Z-Leu-Leu-Leu-al | Sigma-Aldrich (Wisconsin, USA) | Cat#2211 | |
| chemical compound, drug | γ-Secretase inhibitor IX | Calbiochem (California, USA) | Cat#565770 | |
| chemical compound, drug | PhosStop | Roche | Cat#04906837001 | |
| chemical compound, drug | COmplete protease inhibitor cocktail | Roche | Cat#04693132001 | |
| chemical compound, drug | rProtein G Agarose | Invitrogen (Hong Kong) | Cat#15920–010 | |
| chemical compound, drug | DAPI | Sigma-Aldrich (Missouri, USA) | Cat# D9542 | |
| commercial assay or kit | In situ Cell Death Detection Kit, Fluorescein | Roche | Cat#11684795910 | |
| cell line (Human) | HEK 293T cells | ATCC (Virginia, USA) | CRL-3216, RRID: CVCL_0063 | |
| strain, strain background (C57BL/6N) | Mouse: C57BL/6N | Laboratory Animal Unit at the University of Hong Kong | N/A | |
| strain, strain background (C57BL/6N) | Mouse: Sox2-CreER | The Jackson Laboratory (*Arnold et al., 2011*) | RRID:IMSR_JAX:017593 | |
| strain, strain background (C57BL/6N) | Mouse: Pax2-Cre | A. Grove (*Ohyama and Groves, 2004*) | RRID: MMRRC_010569-UNC | |
| strain, strain background (C57BL/6N) | Mouse: B2-r4-Cre | K.S.E. Cheah (*Szeto et al., 2009*) | RRID:MGI:3849737 | |
| strain, strain background (C57BL/6N) | Mouse: Rosa26N1-IC | The Jackson Laboratory (*Murtaugh et al., 2003*) | RRID:IMSR_JAX:008159 | |
| strain, strain background (C57BL/6N) | Mouse: R26REYFP | The Jackson Laboratory (*Srinivas et al., 2001*) | RRID:IMSR_JAX:006148 | |
| strain, strain background (C57BL/6N) | Mouse: Eya1-/- | P.X. Xu (*Xu et al., 1999*) | RRID:MGI:3054666 | |
| strain, strain background (C57BL/6N) | Mouse: Six1-/- | P.X. Xu (*Laclef et al., 2003*) | RRID:MGI:2655196 | |
| sequence-based reagent | Full list of primers for cloning in *Table 2* | N/A | N/A | |
| sequence-based reagent | Full list of primers for genotyping in *Table 1* | N/A | N/A | |
| recombinant DNA reagent | In situ hybridization probe : Dlx5 | (*Liu et al., 1997*) | N/A | |
| recombinant DNA reagent | In situ hybridization probe : Dlx1 | (*Qiu et al., 1995*) | N/A | |
| recombinant DNA reagent | In situ hybridization probe : Eya1 | (*David et al., 2001*) | N/A | |
| recombinant DNA reagent | In situ hybridization probe : Six1 | (*Pandur and Moody, 2000*) | N/A | |
| recombinant DNA reagent | In situ hybridization probe : Sox2 | (*De Robertis et al., 1997*) | N/A | |
| recombinant DNA reagent | In situ hybridization probe : Fgf8 | (*Crossley and Martin, 1995*) | N/A | |
| recombinant DNA reagent | In situ hybridization probe : Fgf3 | (*Wilkinson et al., 1988*) | N/A | |
| recombinant DNA reagent | In situ hybridization probe : Hes1 | (*Zheng et al., 2000*) | N/A | |
| recombinant DNA reagent | In situ hybridization probe: Fgfr1 | Addgene (Massachusetts, USA) | Cat# 14005 | |
| recombinant DNA reagent | In situ hybridization probe: Crabp1 | (IMAGE 2922473) | N/A | |
| recombinant DNA reagent | In situ hybridization probe: Neurog2 | (IMAGE 468821) | N/A | |
| transfected construct (mouse) | pcDNA3.1+ | Addgene | Cat# V790-20 | |

*Continued on next page*

*Continued*

| Reagent type (species) or resource | Designation | Source or reference | Identifiers | Additional information |
|---|---|---|---|---|
| transfected construct (mouse) | pcDNA3.1-Myc-N1ICD | C.C. Hui (Toronto, Canada) | N/A | |
| transfected construct (mouse) | pCS2+-N1DEF-Myc | U. Lendahl (**Chapman et al., 2006**) | N/A | |
| transfected construct (mouse) | pcDNA5-FRT-TO-N1FL-Myc | U. Lendahl (**Chapman et al., 2006**) | N/A | |
| transfected construct (mouse) | P3XFLAG-myc-CMV-26 | Sigma-Aldrich (Missouri, USA) | Cat# E6401 | |
| transfected construct (mouse) | pCMV-3XFlag-N1ICD | This paper | N/A | Generated by cloning the N1ICD fragment into the pCMV-3XFlag-Myc-26 vector between *BamHI* and *EcoRI* sites |
| transfected construct (mouse) | pCMV-3XFlag-N1ICDT2122A | This paper | N/A | Point mutation, primers listed in Table 2 |
| transfected construct (mouse) | pCMV-3XFlag-N1ICDT2122D | This paper | N/A | Point mutation, primers listed in Table 2 |
| transfected construct (mouse) | pCMV-3XFlag-N1ICDT2487A | This paper | N/A | Point mutation, primers listed in **Table 2** |
| transfected construct (mouse) | Flag-Eya1 | P.X. Xu (**Li et al., 2017**) | N/A | |
| transfected construct (mouse) | HA-Eya1 | P.X. Xu (**Li et al., 2017**) | N/A | |
| transfected construct (mouse) | Flag-Eya1D327N | P.X. Xu (**Li et al., 2017**) | N/A | |
| transfected construct (mouse) | Flag-Eya1C | P.X. Xu (**Li et al., 2017**) | N/A | |
| transfected construct (mouse) | Flag-Eya1N | P.X. Xu (**Li et al., 2017**) | N/A | |
| transfected construct (mouse) | Flag-Eya1-DYY | This paper | N/A | Point mutation, primers listed in Table 2 |
| transfected construct (mouse) | Flag-Eya1-Y4 | This paper | N/A | Point mutation, primers listed in Table 2 |
| transfected construct (human) | pCMV-Myc CDC4 WT* | Addgene | Cat# 16652 | |
| transfected construct (human) | HA-Ubiquitin | Addgene | Cat# 18712 | |
| software, algorithm | ImageJ | http://imagej.nih.gov/ij/ | ImageJ, RRID:SCR_003070 | |
| software, algorithm | Prism Version 6 | http://www.graphpad.com | GraphPad Prism, RRID:SCR_002798 | |

## Experimental animals

The following mouse lines were used in this study: wildtype C57BL/6N, $Eya1^{-/-}$ (**Xu et al., 1999**), $Six1^{-/-}$ (**Laclef et al., 2003**), *Pax2-Cre* (**Ohyama and Groves, 2004**), *B2-r4-Ccre* (**Szeto et al., 2009**), *Sox2-CreER* (**Arnold et al., 2011**), $Rosa^{N1-IC}$ (**Murtaugh et al., 2003**) and $R26R^{EYFP}$ (**Srinivas et al., 2001**). Mice were maintained on a C57BL/6N background. Genotyping was conducted by PCR using primers shown in **Table 1**. To activate the CreERT2 protein, tamoxifen (06734, Sigma-Aldrich, Wisconsin, USA) was dissolved in corn oil and administered by intraperitoneal injection to pregnant females using 0.1 mg/g body weight at E7.5 (TM E7.5) and E9.5 (TM E9.5). At least five

embryos were analysed for each genotype at each stage. All mouse experiments were approved by the University of Hong Kong animal research ethics committee (CULATR No. 3329–14 and 3862–15).

## Scanning electron microscopy

Embryos were fixed overnight at 4°C in 2.5% glutaraldehyde in 0.1M phosphate buffer pH 7.4, dehydrated through a graded series of ethanol (30%, 50%, 75%, 95%, 100%), and dried with a critical-point dryer. Embryos were mounted and coated with platinum with sputter coater before analysis and image capturing with a Hitachi S-3400N variable pressure scanning electron microscope. At least three embryos were analysed for each genotype.

## Whole mount in situ hybridization

Mouse embryos were harvested at E8.0, E8.5, E9.0 and E9.5, and fixed in 4% paraformaldehyde (PFA) for overnight at 4°C. After fixation, embryos were dehydrated with a series of methanol/PBST solution. Whole-mount in situ hybridization was performed after rehydration, proteinase K (P6556, Sigma-Aldrich, Missouri, USA) treatment and re-fixation with 4% PFA/0.1% glutaraldehyde. Hybridization with DIG-labeled riboprobes was carried out under high stringency conditions at 60°C. Samples were blocked with 10% blocking reagent and 20% horse serum solution, incubated overnight with anti-Digoxigenin-alkaline phosphatase (11093274910, Roche, Germany). After washing for 24 hr, embryos were incubated with BM purple substrate until clear signal were detected. Probes for *Eya1* (*David et al., 2001*), *Six1* (*Pandur and Moody, 2000*), *Crabp1* (IMAGE468821), *Dlx1* (*Qiu et al., 1995*), *Dlx5* (*Liu et al., 1997*), *Sox2* (*De Robertis et al., 1997*), *Fgf3* (*Wilkinson et al., 1988*), *Fgf8* (*Crossley and Martin, 1995*), *Fgfr1* (14005, Addgene, Massachusetts, USA), *Spry1* (*Minowada et al., 1999*), *Neurog2* (IMAGE 2922473) and *Hes1* (*Zheng et al., 2000*) have been described. Probes for *Brn1*, *DLL1*, *Fgf15* and *Hey1* were generated from C57BL/6N mouse cDNA by PCR with primers shown in Table S2 and then subcloned into pBluescript II KS (+), which was linearized and used as a template for probe synthesis using the restriction enzyme and RNA polymerase. For sectioning of whole-mount in situ hybridization stained embryos, samples were washed in PBS, embedded in gelatin and sectioned at 10 μm. All in situ hybridization experiments were performed on a minimum of 5 embryos for each genotype and for each probe.

## TUNEL assay

TUNEL (terminal deoxynucleotidyl transferase-mediated dUTP-biotin nick end labeling) assay was performed to detect cells under apoptosis. Embryos were harvested, fixed, embedded in paraffin and sectioned. After antigen retrieval, sections were incubated with TUNEL reaction mix 1:10 (In situ Cell Death Detection Kit, 11684795910, Fluorescein, Roche) at 37°C for 30 min. The sections were then washed in PBS at room temperature for 3 times and 5 min each, mounted and images captured using an Olympus fluorescence microscope (BX51). Apoptotic cells were counted blind on three sections per embryo (n = 6). Analysis of variance was performed and significance was estimated using Student's *t*-test.

## Histology and immunohistochemistry

Mouse embryo samples were fixed in 4% PFA overnight or 1 hr 4°C, and embedded in gelatin or dehydrated and embedded in wax. Paraffin and cryo-sections were generated at 6μm and 10 μm, respectively, and subjected to immunohistochemistry. Paraffin section immunostaining was performed using anti-Sox2 (1/500, GT15098-100, Neuromics, Minnesota, USA), anti-Sox3 (1/500, AF2569, R and D systems, Minnesota, USA), anti-Jagged1 (1/300, Ts1.15h, DSHB, Iowa, USA) and anti-E-cadherin (1/1000, 3915, Cell signalling Technology, Massachusetts, USA) antibodies. Cryo-section immunostaining was performed using anti-Eya1(1/300, H00002138-A01, Abnova, Taiwan), anti-Sox2 (1/500, GT15098, Neuromics), anti-Notch1 (1/400, sc-6014, Santa Cruz Biotechnology, Texas, USA), anti-Notch1 ICD (V1744) (1/500, 2421S, Cell Signaling), anti-Notch3 (1/1000, 130502, BioLegend, California, USA), anti-Neurog2 (1/1000, MAB3314, R and D systems), anti-GFP (1/1000, ab6556, abcam, Hong Kong) and anti-Islet1 (1/400, PCRP-ISL1-1A9, DSHB), followed by incubation with Alexa Fluor 488 donkey anti-goat IgG (1/500, A-11055, Thermo Fisher Scientific, Hong Kong), Alexa Fluor 488 donkey anti-rabbit IgG (1/500, A-21206, Thermo Fisher Scientific), Alexa Fluor 488 donkey anti-rat IgG (1/500, A-21208, Thermo Fisher Scientific), Alexa Fluor 555 donkey anti-mouse

IgG (1/500, A-31570, Thermo Fisher Scientific), or Alexa Fluor 594 donkey anti-goat IgG (1/500, A-11058, Thermo Fisher Scientific). Sections were counterstained with DAPI (D9542, Sigma-Aldrich, Missouri, USA) prior to mounting. Sections from at least three different embryos were stained with each antibody.

## Cell transfections and DNA constructs

HEK 293T cells (CRL-3216, ATCC, Virginia, USA) were tested negative for mycoplasma contamination by using the universal mycoplasma detection kit (ATCC 30–1012K). Cells were maintained in DMEM medium supplemented with 10% FBS at 37°C with 5% $CO_2$. Transient transfection of 293 T cells was performed using X-treme 9 DNA Transfection Reagent (6365779001, Roche). The following constructs were used for overexpression: *pcDNA3.1-Myc-N1ICD* (from C.C. Hui), *pCS2+-N1ΔEF-Myc* and *pcDNA5-FRT-TO-N1FL-Myc* (*Chapman et al., 2006*), *Flag-Eya1, HA-Eya1, Flag-Eya1D327N, Flag-Eya1C* and *Flag-Eya1N* (from P.X. Xu), *Flag-Eya1-DYY, Flag-Eya1-Y4,* pCMV-*3XFlag-N1ICD,* pCMV-*3XFlag-T2122A,* pCMV-*3XFlag-T2122D, pCMV-3XFlag-T2487A, pCMV-myc-Fbw7* and *HA-Ubiquitin* (18712, Addgene). *pCMV-3XFlag-N1ICD* was generated by cloning *the N1ICD* fragment into the *pCMV-3XFlag-Myc-26* vector between *BamHI* and *EcoRI* sites. *Flag-Eya1-DYY, Flag-Eya1-Y4,* pCMV-*3XFlag-T2122A,* pCMV-*3XFlag-T2122D* and pCMV-*3XFlag-T2487A* were generated by site-directed mutagenesis and confirmed by DNA sequencing (primer sequences shown in *Table 2*).

## Drug treatment

Transfected HEK 293T cells were cultured in normal DMEM with 10% FBS and penicillin/streptomycin. Cycloheximide (C7698, Sigma-Aldrich, Missouri, USA) was added into the medium at a final concentration of 200 µM to stop protein synthesis for indicated time before western blot analysis. For protein ubiquitination analysis, MG132 (2211, Sigma-Aldrich, Wisconsin, USA) was added into the medium at 20 µM for 6 hr to stop protein degradation. To inhibit γ-secretase activity, 12 hr before transfection, DAPT (565770, Calbiochem, California, USA) was added to the medium at 10 µM. After transfection, DAPT was added to the medium for 48 hr prior to harvesting the cells.

## SDS-PAGE and western blot analysis

Cells or dissected embryo tissues were harvested with lysis buffer containing protease inhibitor (04693132001, Roche) and phosphatase inhibitor (PhosStop, 04906837001, Roche). Cells were lysed by incubating in lysis buffer at 4°C for 20 min. The total protein concentration was measured by Protein BCA assay. Before SDS-PAGE, the protein lysate was denatured by boiling with SDS loading buffer for 5 min. SDS-PAGE polyacrylamide gels of various percentages from 6% to 10% were used. After protein transfer, the nitrocellulose membrane was blocked with 5% milk powder at room temperature for 1 hr. Primary antibodies were diluted in 5% BSA solution. Antibodies for western blot were: anti-Notch1 (1/300, sc-6014, Santa Cruz), anti-Eya1 (1/500, H00002138-A01, Abnova), anti-Hes1 (1/500, ab71559, Abcam), anti-Hey1 (1/500, ab22614, Abcam), anti-Gapdh (1/10000, ab8245, Abcam), anti-Notch1 1ICD (V1744) (1/500, 2421S, Cell Signaling), anti-Flag (1/4000, F7425, Sigma-Aldrich, Missouri, USA), anti-Myc (9E10) (1/1000, sc-40, Santa Cruz), anti-α-tubulin (1/5000, AA4.3-s, DSHB), anti-HA (1/1000, sc-805, Santa Cruz), anti-Phospho-Threonine-Proline (1/3000, 9391S, Cell Signaling), anti-Phospho-Serine-Proline-Proline motif (1/1000, 14390, Cell Signaling), and anti-Phospho-Tyrosine (1/1000, PY99, Santa Cruz). The membrane and primary antibody solution were incubated at 4°C for overnight. Then the membrane was washed and incubated with secondary antibodies, Donkey anti-goat HRP Conjugate (1/5000, sc-2020, Santa Cruz), Donkey anti-Mouse HRP (1/5000, A16017, Thermo Fisher), Donkey anti-Rabbit (1/5000, A16035, Thermo Fisher) with 5% BSA at room temperature for 1 hr. After incubation and washing, the chemiluminescent substrate (Thermo Scientific Pierce) was added and signals recorded by film or Gel Doc system (Bio-Rad).

## Immunoprecipitation

For immunoprecipitation (IP) assay, transfected HEK 293T cells were lysed in lysis buffer (50 mM Tris-HCl (pH7.4) containing 1.0% Nonidet P-40, 150 Mm NaCl, 5 mM EDTA, protease inhibitor (04693132001, Roche) and phosphatase inhibitor (PhosStop, 04906837001, Roche), then spun and the supernatant was collected for IP. IP analysis was performed by incubating the supernatant with anti-Myc (9E10) (sc-40, Santa Cruz) or anti-FLAG M2 (F1804, Sigma-Aldrich, Missouri, USA)

antibodies at 4°C for 1 hr. Then recombinant Protein G Agarose (15920–010, Invitrogen, Hong Kong) was added into the mixture and incubated at 4°C for 1 hr. After incubation, rProtein G Agarose was washed with lysis buffer three times. The precipitated proteins were collected by boiling the rProtein G Agarose with loading buffer.

### Protein purification

The Flag-tagged proteins were purified from transfected 293T cells using anti-FLAG M2 (F1804, Sigma-Aldrich) antibody and rProtein G Agarose. The 293T cells were lysed in RIPA buffer with protease inhibitor (04693132001, Roche) and phosphatase inhibitor (PhosStop, 04906837001, Roche) at 4°C for 20 min. After centrifugation at 12,000 rpm for 20 min, the supernatant was incubated with anti-FLAG M2 antibodies and rProtein G Agarose with protease and phosphatase inhibitors at 4°C overnight. After incubation, the rProtein G Agarose was washed with RIPA buffer three times, and the protein was eluted with 50 mM Tris-HCl buffer (pH 7.5) containing 150 mM NaCl, 10% glycerol and 100 µg/ml of 3XFLAG peptide (F4799, Sigma). The eluted proteins were confirmed by the Coomassie staining method and were collected for in vitro phosphatase assay.

### In vitro phosphatase assay

Purified Flag-Eya1 and Flag-N1ICD or Flag-T2122A were incubated in the reaction buffer (50 mM Tris-HCl buffer (pH 7.5), 25 mM $MgCl_2$, 0.5 mM EDTA, and 1 mM DTT) at 37°C for 1 hr. The reaction were stopped and analyzed on 6% SDS-PAGE. Western blot analyses were performed with the anti-phosphopeptide-specific antibodies for threonine, serine and tyrosine and anti-FLAG antibodies.

### Quatification and statistical analysis

Protein levels were estimated by ImageJ. Data were presented as means ± standard errors of the means (SEMs). Graphs were generated and statistical analyses were performed using Prism. The relative protein level in Notch1 ICD stability assay was fitted with the one-phase decay. To quantify the number of Sox2[+] cells in the geniculate placode, green fluorescent positive cells on consecutive coronal sections of E9.5 wildtype (17 ± 1 sections/embryo) and Eya1[-/-] embryos (7 ± 1 sections/embryo) were counted (n = 3 embryos). Statistical significance between two groups was determined by two-tailed Student's t-test. Statistically significant differences are: *p<0.05, **p<0.01 and ***p<0.001.

## Acknowledgements

We would like to thank CC Hui for providing the Notch1 ICD plasmid; J Tanner, R Kok, DY Jin and CM Qian for technical advice. This work was supported by research grants from the Hong Kong Research Grants Council (RGC GRF 777411, RGC GRF 17113415) to MHS; and research grants from the Swedish Research Council (K2014-64X-20097-09-5), the Swedish Cancer Society (CAN 2016/271) and ICMC (Integrated Cardio Metabolic Center) to UL; and NIH RO1 grants (DK064640 and DC014718) to PXX.

## Additional information

### Funding

| Funder | Grant reference number | Author |
| --- | --- | --- |
| National Institutes of Health | DK064640 | Pin-Xian Xu |
| National Institutes of Health | DC014718 | Pin-Xian Xu |
| Vetenskapsrådet | K2014-64X-20097-09-5 | Urban Lendahl |
| Cancerfonden | CAN 2016/271 | Urban Lendahl |
| Integrated Cardio Metabolic Center | | Urban Lendahl |
| Research Grants Council, University Grants Committee | RGC GRF 777411 | Mai Har Sham |

| Research Grants Council, University Grants Committee | RGC GRF 17113415 | Mai Har Sham |

The funders had no role in study design, data collection and interpretation, or the decision to submit the work for publication.

## Author contributions

Haoran Zhang, Conceptualization, Formal analysis, Validation, Investigation, Visualization, Methodology, Writing—original draft; Li Wang, Formal analysis, Validation, Investigation, Visualization, Methodology, Writing—original draft; Elaine Yee Man Wong, Resources, Supervision, Methodology; Sze Lan Tsang, Supervision, Methodology; Pin-Xian Xu, Resources, Writing—review and editing; Urban Lendahl, Formal analysis, Writing—review and editing; Mai Har Sham, Conceptualization, Resources, Formal analysis, Supervision, Funding acquisition, Project administration, Writing—review and editing

## Author ORCIDs

Mai Har Sham (iD) http://orcid.org/0000-0003-1179-7839

## Ethics

Animal experimentation: All mouse experiments were performed in strict accordance with the recommendations and approved by the University of Hong Kong animal research ethics committee (CULATR No. 3329-14 and 3862-15).

## Decision letter and Author response

Decision letter https://doi.org/10.7554/eLife.30126.030
Author response https://doi.org/10.7554/eLife.30126.031

# Additional files

## Supplementary files

• Transparent reporting form
DOI: https://doi.org/10.7554/eLife.30126.029

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
