## [Decision Letter]

Thank you for submitting your article "An Eya1-Notch axis specifies neurogenic and non-neuronal epibranchial placodes for craniofacial morphogenesis" for consideration by *eLife*. Your article has been reviewed by three peer reviewers, and the evaluation has been overseen by Marianne Bronner as the Senior and Reviewing Editor. The following individual involved in review of your submission has agreed to reveal their identity: Jean-Pierre Saint-Jeannet (Reviewer #3).

The reviewers have discussed the reviews with one another and the Senior Editor has drafted this decision to help you prepare a revised submission. We have included the full reviews since there was some discrepancy between the opinions of the reviewers and we felt this might be helpful.

Summary:

The authors investigate the role of Eya1 in epibranchial and pharyngeal arch development in the mouse. Using Eya1 knock out mice combined with marker analysis and lineage tracing, the results suggest that epibranchial placodes generate two different cell populations: neurons and non-neuronal cells both of which arise from Sox2^+^ progenitors. The authors find that Notch signaling is downstream of Eya1 and provides a molecular mechanism in which Eya1 phosphatase activity regulates Notc1 ICd stability.

It has been assumed that some cells in the epibranchial placodes do not form neurons, and some of these non-neurogenic domains were known. However, this is the first study to use multiple markers to explore this region and demonstrates a unique genetic requirement for its integrity. Unfortunately, their model the authors present has a number of big holes, and even their marker characterization of the non-neural domain ignores some of the gene expression that their own figures show. The paper needs to provide a clearer sense of what is actually going on in this non-neurogenic region – what/where is the relevant Notch ligand, as well as reconcile their model with known data (e.g. the lack of a pharyngeal phenotype in FGF3 nulls).

Essential revisions:

1) The authors need to show additional genotypes from their crosses in some of the figures. They should have these mice already, so additional images of these should be included.

2) What is the ligand that is activating Notch1 during the decision between Ngn2-expressing neurogenic placode tissue and Sox2/FGF3-expressing non-neural ectoderm? Dll1 is only expressed in delaminating neuroblasts – well away from the pharyngeal ectoderm – and Jag1 is only expressed in the pouch region. Either there is another Notch ligand responsible for activation of the Hes/Hey genes in the arch ectoderm, or else another signal (such as FGF3 or 8) may be activating these genes. Moreover, Sox2 and FGF3 only label the anterior ("rostral-proximal pharyngeal" as defined by the authors) ectoderm of each arch (Figure 2), yet the Notch target genes Hes1 and Hey1 are expressed throughout the arch ectoderm (Figure 4). Finally, there appears to be very little Notch-ICD staining in the rostral-proximal pharyngeal compared to the arch mesenchyme adjacent to it, both in the wild type embryos and mutants (Figure 4' and D, D'). This may again indicate that other signals might be activating the Hes/Hey genes in this region.

3) There is some confusion in the results between whether Eya1 is modifying and regulating the cleaved Notch1 ICD and the intact Notch1 receptor. Figure 4 shows that the amount of Notch C20 staining decreases dramatically in the Eya1 mutants, and the authors conclude this suggests "that the total amount of Notch1 receptor was reduced in the ectodermal cells rather than that receptor cleavage was affected." However, in Figure 6 and Figure 7, the focus is on Eya1 modification and stabilization of the cleaved form. Is it possible that Eya1's threonine phosphatase activity regulates the stability of both the intact Notch1 receptor and its ICD?

4) In the rescue experiment in Figure 5, the over-expression of Notch-ICD activates Sox2 in the entire arch ectoderm (Figure 5) but not in the Eya1 mutant (Figure 5). This does not fit in with the model of Eya1-stabilized Notch signaling promoting Sox2 expression in the rostral-proximal pharyngeal ectoderm. The authors need to consider/explain this result, as they don't mention it at all in the text. Part of the problem in interpreting this experiment is that the authors need to show wild type and *Eya1^-/-^* embryos as well as the *Pax2-Cre*; ROSA-N1ICD and *Pax2-Cre*; ROSA-N1ICD;*Eya1^-/-^* images in Figure 5.

5) The biggest issue is that the model has many holes, and the authors need to figure out a way of either plugging the holes or simplifying the model. They should provide a clearer sense of what is going on in this non-neurogenic region – what/where is the relevant Notch ligand, etc., as well as reconcile their model with known data (e.g. the lack of a pharyngeal phenotype in FGF3 nulls).

6) In the Discussion, the authors attribute their arch phenotype to a loss of FGF3: "Therefore, one can assume that phenotypes observed only in *Eya1^-/-^* embryos can be attributed to the non-neuronal epibranchial placode, and more specifically to the lack of *Fgf3* expression" and: "The failure to organize the PA architecture in the *Eya1^-/-^* embryos are likely due to defective FGF signaling". FGF3 mutant mice do not have a clear arch phenotype. Ectoderm-specific FGF8 mutants have an arch phenotype that is much more severe than Eya1 mutants. The authors need to change these parts of the discussion as it impacts their model.

*Reviewer #1:*

This study investigates the role of Eya1 in epibranchial and pharyngeal arch development in mouse. In particular, the authors use Eya1 knock out mice combined with marker analysis, lineage tracing to suggest that epibranchial placodes generate two different cell populations, neurons and non-neuronal cells. They claim that both arise from Sox2^+^ progenitors, and that the non-neural cells expressing various FGF ligands, are a so far not discovered cell population and act as a signaling centre to patter the pharyngeal arches. They show that Notch signaling, a mediator of neurogenesis, is downstream of Eya1 and provide a molecular mechanism in which Eya1 phosphatase activity regulates Notc1 ICd stability. Overall, they make three claims: i) discovery of a new cell population, the non-neural derivatives of epibranchial placodes; ii) the signaling centre activity of these cells with FGFs as mediators; iii) Notch activity regulation by Eya1 through regulating phosphorylation, NICD stability/degradation.

The strongest part of the paper are the biochemical experiments presented in Figure 6 and Figure 7, which provide new insight into how Eya1 interacts with the Notch pathway, how Notch signaling is regulated and the integration of key factors in placode development (Eya1) and downstream signaling. Thus, the evidence presented for the 3rd claim is solid and novel.

However, the other two points are do not provide any novelty and / or are not supported by the authors' data.

1) The authors argue that they have uncovered a 'novel lineage-related non-neural' derivative of epibranchial placodes, in addition to the neurogenic cells. Epibranchial placodes are focal thickenings, which in a Notch-Delta lateral inhibition mediated manner generate neurons. To the best of my knowledge there is no claim in the literature that *all* epibranchial cells form neurons, but rather that all are competent and only some do, and this is evidenced by studies showing individual delaminating cells. So, the authors do not discover a new cell population that is derived from Sox2^+^ progenitor cells, but simply describe those cells that remain in the ectoderm. This is not a novel finding, but seems to give a name to cells that are known to exist.

They claim that this novel domain is Sox2/Fgf3 positive, while the neurogenic domain is Sox2/Ngn2 +. However, looking at the data provided in Figure 3 this does not seem to be the case and FGF3 expression appears to encompass both Ngn2 + and – regions (see below), contradicting their notion that these are two different regions or cells with different identities.

2) The authors claim that these cells act as an FGF3^+^ signaling centre important for pharyngeal arch development. First, they do not provide any evidence that this region acts as a signaling centre beyond showing that it expresses various FGF ligands, and second the fact that FGF activity is important in this context from the ectoderm as well as from the endoderm is already known (e.g. findings from the Partanen, Scambler, Basson, Kimmel groups). Thus, this finding is not new.

I therefore feel that while the Eya1-Notch connection is new and very nicely demonstrated, the paper does not move the field forward in a major way by providing new concepts or being highly influential in a broader sense.

*Reviewer #2:*

Eya1 is known to be a transcriptional co-activator of Six genes, but the fact that (1) Eya proteins do not bind DNA (2) some have phosphatase activities and (3) the phenotypes of Eya and Six mutant genes expressed in the same domain can differ suggests that Eya proteins may have additional roles. Here, the authors suggest that Eya1's threonine phosphatase activity can modify Notch, enhancing the stability of the Notch1 intracellular domain. They suggest that this may play a role in the differentiation of pharyngeal ectoderm derived from a non-neurogenic region of the epibranchial placodes.

The data convincingly shows that:

- Six1 and Eya1 mutants differ in their phanrygeal arch and epibranchial placode phenotypes.

- That Sox2 and FGF3 end up defining a non-neurogenic region of epibranchial placode epithelium, although Ngn2 cells initially delaminate from this domain.

- Notch1 receptor expression and Notch signaling are reduced in Eya1 mutants, and Notch1-ICD over-expression can rescue aspects of the Eya1 arch phenotype.

- Phosphorylation of the Notch1 ICD by Eya1's threonine phosphatase activity stabilizes the ICD, allowing persistence of the Notch signal.

The authors present a model in which Eya1 titrates the strength/persistence of Notch signaling, which regulates a choice between producing neurogenic placodal ectoderm (Ngn2+) and non-neural pharyngeal ectoderm (Sox2/FGF3^+^).

Although the data is convincing, the interpretation of the data and model have some weaknesses.

1) What is the ligand that is activating Notch1 during the decision between Ngn2-expressing neurogenic placode tissue and Sox2/FGF3-expressing non-neural ectoderm? Dll1 is only expressed in delaminating neuroblasts – well away from the pharyngeal ectoderm – and Jag1 is only expressed in the pouch region. Either there is another Notch ligand responsible for activation of the Hes/Hey genes in the arch ectoderm, or else another signal (such as FGF3 or 8) may be activating these genes. Moreover, Sox2 and FGF3 only label the anterior ("rostral-proximal pharyngeal" as defined by the authors) ectoderm of each arch (Figure 2), yet the Notch target genes Hes1 and Hey1 are expressed throughout the arch ectoderm (Figure 4). Finally, there appears to be very little Notch-ICD staining in the rostral-proximal pharyngeal compared to the arch mesenchyme adjacent to it, both in the wild type embryos and mutants (Figure 4' and D, D'). This may again indicate that other signals might be activating the Hes/Hey genes in this region.

2) There is some confusion in the results between whether Eya1 is modifying and regulating the cleaved Notch1 ICD and the intact Notch1 receptor. Figure 4 shows that the amount of Notch C20 staining decreases dramatically in the Eya1 mutants, and the authors conclude this suggests "that the total amount of Notch1 receptor was reduced in the ectodermal cells rather than that receptor cleavage was affected." However, in Figure 6 and Figure 7, the focus is on Eya1 modification and stabilization of the cleaved form. Is it possible that Eya1's threonine phosphatase activity regulates the stability of both the intact Notch1 receptor and its ICD?

3) In the rescue experiment in Figure 5, the over-expression of Notch-ICD activates *Sox2* in the entire arch ectoderm (Figure 5) but not in the Eya1 mutant (Figure 5). This does not fit in with the model of Eya1-stabilized Notch signaling promoting Sox2 expression in the rostral-proximal pharyngeal ectoderm. The authors need to consider/explain this result, as they don't mention it at all in the text. Part of the problem in interpreting this experiment is that the authors need to show wild type and *Eya1^-/-^* embryos as well as the *Pax2-Cre*; ROSA-N1ICD and *Pax2-Cre*; ROSA-N1ICD;*Eya1^-/-^* images in Figure 5.

4) In the Discussion, the authors attribute their arch phenotype to a loss of FGF3: "Therefore, one can assume that phenotypes observed only in *Eya1^-/-^* embryos can be attributed to the non-neuronal epibranchial placode, and more specifically to the lack of Fgf3 expression" and: "The failure to organize the PA architecture in the *Eya1^-/-^* embryos are likely due to defective FGF signaling". FGF3 mutant mice do not have a clear arch phenotype. Ectoderm-specific FGF8 mutants have an arch phenotype that is much more severe than Eya1 mutants. The authors need to change these parts of the discussion as it impacts their model.

In sum, I think the data of the paper is very solid, but at the moment I think their interpretation and model is incomplete.

*Reviewer #3:*

Eya1 is an important regulator of cranial placode formation typically acting as a Six1 cofactor. In this manuscript, Zhang and colleagues describe the unique function of Eya1 in the development of epibranchial placodes and proximal pharyngeal arches (PA). Unlike Six1 mutants, Eya1 deficient embryos exhibit hypoplastic 2nd and 3rd PA, defective pharyngeal clefts and pouches formation, and failure to express Ngn2 in epibranchial placode cells. The PA phenotype can be traced back to a previously unrecognized non-neuronal lineage derived from the epibranchial placode region and acting as a signaling center to pattern the PA. The authors provide evidence that Eya1 directs the development of this non-neuronal epibranchial placode lineage through Notch signaling by promoting Notch1 ICD stabilization via Eya1 phosphatase activity.

The manuscript makes an important contribution to the field. The overall quality of the work is superb, combining mouse genetics and biochemical analyses. The manuscript is well presented and beautifully illustrated – it is likely to become a "classic". The link between the phosphatase activity of Eya1 and Notch stability/activity in the context of epibranchial placode formation is both novel and very compelling.

The data in support of the authors' conclusions are strong:

-Lineage analyses indicate that the epibranchial placode can be subdivided into 2 distinct Sox2^+^ lineages, one neurogenic (Ngn2+) and the other non-neurogenic (Sox2^+^ and Fgf3^+^). The later contributes to the external auditory canal and pinna epithelium, further confirming that these cells are non-neuronal in nature.

-The rescue of the PA phenotype and Fgf3 expression domain in the PA ectoderm of Eya1 mutants by expression of Notch ICD makes a very convincing case for an Eya1-Notch nexus in the regulation of epibranchial placode and proximal PA morphogenesis.

-The biochemical data (Figure 6) convincingly demonstrate that Eya1 interacts with Notch1 ICD via its C-terminal domain and acts as a threonine 2122 phosphatase to control Notch1 ICD stability, thereby providing a molecular mechanism for the regulation of cell fate in this placodal region.

---

## [Author Response]

Essential revisions:1) The authors need to show additional genotypes from their crosses in some of the figures. They should have these mice already, so additional images of these should be included.

We appreciate this comment, for space reasons we had left out data from some control genotypes or which appeared in other figures. To address this, we now include data from wildtype and *Eya1^-/-^* mice in Figure 5 (new Figure 5, and 5X; see also response to reviewer 1 below). We also made cross-references to other control data in the manuscript.

2) What is the ligand that is activating Notch1 during the decision between Ngn2-expressing neurogenic placode tissue and Sox2/FGF3-expressing non-neural ectoderm? Dll1 is only expressed in delaminating neuroblasts – well away from the pharyngeal ectoderm – and Jag1 is only expressed in the pouch region. Either there is another Notch ligand responsible for activation of the Hes/Hey genes in the arch ectoderm, or else another signal (such as FGF3 or 8) may be activating these genes. Moreover, Sox2 and FGF3 only label the anterior ("rostral-proximal pharyngeal" as defined by the authors) ectoderm of each arch (Figure 2), yet the Notch target genes Hes1 and Hey1 are expressed throughout the arch ectoderm (Figure 4). Finally, there appears to be very little Notch-ICD staining in the rostral-proximal pharyngeal compared to the arch mesenchyme adjacent to it, both in the wild type embryos and mutants (Figure 4' and D, D'). This may again indicate that other signals might be activating the Hes/Hey genes in this region.

This is a relevant and interesting question on Notch signaling factors, which we agree was not clearly addressed in our original version. We have now provided expression data of the Notch ligands *Dll1* and *Jag1* at earlier stages (Figure 4—figure supplement 1). We show that *Dll1* is expressed in the Ngn2-expressing neurogenic placode. *Jag1* is in fact expressed broadly in the epibranchial placode, overlapping with Notch1 and both the Ngn2+ and Sox2^+^/FGF3^+^ cell populations, suggesting that it indeed is the ligand at work during the lineage decisions.

We have also examined the expression of *Dll3* and *Jag2*, but found that neither of them are expressed in the epibranchial placodal region; though Jag2 is expressed in the distal pharyngeal ectoderm (data not shown). We have not examined *Dll4*, as based on the literature (Duarte et al., 2004; Sacilotto et al., 2013), *Dll4* is not expressed in the pharyngeal ectoderm.

The reviewer also raises the important question as to whether Hes1 and Hey1 are (at least in part) regulated by other factors. We believe this is the case, in particular for Hes1. When we revisit our data in the light of this comment, we agree that while Hes1 is downregulated in the epithelium (Figure 4), the western blot data (Figure 4) indicate that Hes1 is not subjected to a robust downregulation in the wider region of the pharyngeal arch tissue used for the Western blot experiment. This suggests that other factors may regulate Hes1 in the pharyngeal arches, for example Hes1 acts downstream of Tbx1, which is expressed in the mesoderm of the pharyngeal arches (van Bueren et al., 2010). This line of reasoning is also in agreement with the original paper on the CSL (RBPJk) knockout mouse (de la Pompa et al., 1997), which shows that Hes1 expression is not downregulated upon ablation of CSL or Notch1 in E9.0 mouse embryos. We have now modified the in the last paragraph of the subsection “Notch signaling is reduced in the pharyngeal epithelium of *Eya1^-/-^* embryos” to better reflect the actual data.

Hey1, on the other hand, shows expression patterns in the *Eya1* mutant and Notch1 ICD gain-of-function experiments, which are consistent with being a bona fide Notch downstream gene in this context. We have rewritten the text in the revised version to better reflect the data on ligand expression and the roles of *Hes* and *Hey* genes as reading out Notch signaling (Results subsection “Notch signaling is reduced in the pharyngeal epithelium of Eya1^-/-^ embryos”, and Figure 5—figure supplement 1).

3) There is some confusion in the results between whether Eya1 is modifying and regulating the cleaved Notch1 ICD and the intact Notch1 receptor. Figure 4 shows that the amount of Notch C20 staining decreases dramatically in the Eya1 mutants, and the authors conclude this suggests "that the total amount of Notch1 receptor was reduced in the ectodermal cells rather than that receptor cleavage was affected." However, in Figure 6 and Figure 7, the focus is on Eya1 modification and stabilization of the cleaved form. Is it possible that Eya1's threonine phosphatase activity regulates the stability of both the intact Notch1 receptor and its ICD?

This is a good point, and our experiments on Notch1 ICD construct alone did not allow us to unambiguously demonstrate whether regulation occurs at the full receptor or ICD levels. To address this, we have conducted new cell transfection experiments, using both full length receptor constructs as well as a truncated but membrane-tethered form of Notch1 (Notch1ΔE). The results show that Eya1 affects not only Notch1 ICD, but also full length and Notch1ΔE. Technically, we blocked γ-secretase cleavage (DAPT) prior to Eya1 addition, to make sure that we primarily measured full length Notch1 and Notch1ΔE proteins that were “locked” in the membrane prior to S3 cleavage. These new data are presented in the new Figure 6 and are discussed at: subsection “Eya1 positively regulates Notch signaling via dephosphorylation of Notch1 ICD”.

4) In the rescue experiment in Figure 5, the over-expression of Notch-ICD activates Sox2 in the entire arch ectoderm (Figure 5) but not in the Eya1 mutant (Figure 5). This does not fit in with the model of Eya1-stabilized Notch signaling promoting Sox2 expression in the rostral-proximal pharyngeal ectoderm. The authors need to consider/explain this result, as they don't mention it at all in the text. Part of the problem in interpreting this experiment is that the authors need to show wild type and Eya1^-/-^ embryos as well as the Pax2-Cre; ROSA-N1ICD and Pax2-Cre; ROSA-N1ICD;Eya1^-/-^ images in Figure 5.

We thank the reviewer for noting this discrepancy between Sox2 regulation by Notch in the two genotypes. In the original version, the Sox2 immunostaining was indeed barely visible in the Pax2-Cre; ROSA-N1ICD;Eya1^-/-^ embryo. We have now performed these experiments in additional embryos (n=6) and we can safely conclude that there is indeed Sox2 activation in the entire pharyngeal epithelium in both wildtype and *Eya1^-/-^* (new Figure 5’). However, we agree that while Notch1 ICD expression increased Sox2 expression in *Eya1^-/-^* embryos, the *Sox2* expression did not reach the level observed in wildtype embryos (compare Figure 5 with 5Q). This is now more clearly stated in the text (subsection “Expression of Notch1 ICD rescues the non-neuronal epibranchial lineage and pharyngeal arch segmentation phenotype in *Eya1^-/-^* embryos”).

The data from E9.5 wildtype and *Eya1^-/-^* embryos are presented in Figure 1 and Figure 2 (for comparison with *Six1^-/-^*), but we agree that it would be important for the interpretation of the data in Figure 5 to show controls (wildtype and *Eya1^-/-^* images) from the same crossings side-by-side. We have therefore revised Figure 5 to now include wildtype and *Eya1^-/-^* data to facilitate comparison (new Figure 5). The text is revised accordingly (see the aforementioned subsection).

5) The biggest issue is that the model has many holes, and the authors need to figure out a way of either plugging the holes or simplifying the model. They should provide a clearer sense of what is going on in this non-neurogenic region – what/where is the relevant Notch ligand, etc., as well as reconcile their model with known data (e.g. the lack of a pharyngeal phenotype in FGF3 nulls).

These are important points, and we agree that our original model in some regards fell short of summarizing the actual data. We have addressed this concern by both providing additional data to support the model, but also tone down certain aspects of the model. To strengthen the model, we have studied the mode of Notch activation further (i.e. which ligands and downstream genes that are activated, see above), and we also provided additional data that FGF signaling indeed is activated downstream of Notch. We agree with the reviewer that Fgf3 is not necessarily the mediator, based on the Fgf3^-/-^ phenotype and redundant functions with other Fgfs that are co-expressed in the pharyngeal region. FGF signaling is however important for epibranchial placode and pharyngeal arch development, as mutations in Fgfr1 and Fgf8 lead to dysregulated development (Trokovic et al., 2003; Abu-Issa et al., 2002; Trumpp et al., 1999; Tucker et al., 1999), and in the revised version, we show that Spry1 (Sprouty1; which is a proxy for downstream FGF signaling) is reduced in the *Eya1^-/-^* embryos, and that Spry1 expression is restored upon Notch1 ICD expression in an *Eya1^-/-^* background, arguing that Notch regulates FGF signaling output independent of Eya1. Importantly, this is not a consequence of altered FGF receptor expression levels, as Fgfr1 expression is not affected by Eya1. With regard to toning down the model, we have omitted the reference to the signaling center in the original model (and in the Discussion), as we admittedly cannot yet tell which FGF (or other morphogens) that may be at work. As also discussed with reviewer 1, we have also toned down the discussion on the non-neuronal cells as a new placode and we now refer to them as “non-neuronal cell population” in Figure 8 (and the text). We would however like to keep the schematic drawing of the molecular mechanism linking Eya1 and Notch, as we believe this is a central part of the paper, and also well supported by data. By providing additional data and making these amendments to the model, we feel that the revised model is not overstretching the data, but summarizes the key findings in a succinct way.

6) In the Discussion, the authors attribute their arch phenotype to a loss of FGF3: "Therefore, one can assume that phenotypes observed only in Eya1^-/-^ 527 embryos can be attributed to the non-neuronal epibranchial placode, and more specifically to the lack of Fgf3 expression" and: "The failure to organize the PA architecture in the Eya1^-/-^ 547 embryos are likely due to defective FGF signaling". FGF3 mutant mice do not have a clear arch phenotype. Ectoderm-specific FGF8 mutants have an arch phenotype that is much more severe than Eya1 mutants. The authors need to change these parts of the discussion as it impacts their model.

We agree that our assumption of *Fgf3* as the signal mediator was premature, for the reasons listed above. As discussed above, we however believe that the upregulation of *Spry1* expression by Notch ICD demonstrates that Notch acts upstream of FGF signaling (new Figure 2’). We have modified the text accordingly, and have removed the discussion of Fgf3 as a potential morphogen and the Fgf3-expressing cell cluster as a potential signaling centre. The model in Figure 8 has also been modified and toned down in accordance with these changes (see point 5 above).

Reviewer #1:[…] 1) The authors argue that they have uncovered a 'novel lineage-related non-neural' derivative of epibranchial placodes, in addition to the neurogenic cells. Epibranchial placodes are focal thickenings, which in a Notch-Delta lateral inhibition mediated manner generate neurons. To the best of my knowledge there is no claim in the literature that all epibranchial cells form neurons, but rather that all are competent and only some do, and this is evidenced by studies showing individual delaminating cells. So, the authors do not discover a new cell population that is derived from Sox2^+^ progenitor cells, but simply describe those cells that remain in the ectoderm. This is not a novel finding, but seems to give a name to cells that are known to exist.

We accept the critique and agree that there is precedence for non-neuronal cells, but that the current literature has not addressed the fates of the epibranchial placodal cells comprehensively. We have now cited examples of non-neuronal cells. Notably, and as also discussed in Essential revisions above, we include references regarding non-classically defined epibranchial placode populations in chick (Abu-Elmagd et al., 2001; Ishii et al., 2001; Tripathi et al., 2009). We have amended the text to hopefully present a more balanced discussion of this topic (subsection “The discovery of a non-neuronal epibranchial cell population reveals a bipotential differentiation program in epibranchial placodes”). In the light of including the earlier literature on non-neuronal differentiation, we have decided to not refer to the non-neuronal cells as a novel type of placode, but as a lineage-related non-neuronal cell population from the epibranchial placodes. The text has been changed in accordance with this, both in the Title, Abstract, Introduction, Results and Discussion as well as in the summary figure (Figure 8). We however still believe that we add to the literature with regard to cell lineages from the epibranchial placodes, and that our study improves our understanding of the molecular basis for lineage segregation.

They claim that this novel domain is Sox2/Fgf3 positive, while the neurogenic domain is Sox2/Ngn2 +. However, looking at the data provided in Figure 3 this does not seem to be the case and FGF3 expression appears to encompass both Ngn2 + and – regions (see below), contradicting their notion that these are two different regions or cells with different identities.

We understand and appreciate this comment. We have now provided better whole mount and matched section data to show the expression of the three genes, but still the visualization of combinations of Sox2/Ngn2/Fgf3 is complicated by the fact that there are no good antibodies to Fgf3, which therefore is analyzed by in situ hybridization for Fgf3, combined with IHC for Ngn2 and Sox2. We agree with the reviewer that at 8-15ss, the Sox2^+^/Fgf3^+^ domain overlaps with the Ngn2+ region. We believe the gradual restriction of the Ngn2+ and Sox2^+^/Fgf3+ cell populations by E9.5 (25ss) is quite clear. In the light of our new analysis, we have modified the text (subsection “Bipotential epibranchial placode differentiation”, first paragraph) and the summary in Figure 3.

2) The authors claim that these cells act as an FGF3^+^ signaling centre important for pharyngeal arch development. First, they do not provide any evidence that this region acts as a signaling centre beyond showing that it expresses various FGF ligands, and second the fact that FGF activity is important in this context from the ectoderm as well as from the endoderm is already known (e.g. findings from the Partanen, Scambler, Basson, Kimmel groups). Thus, this finding is not new.

We understand these comments, and agree that our original suggestion of *Fgf3*^+^ signaling centre was premature, as discussed above. We also agree that we did not do a good job in citing the literature on FGF signaling. As discussed above, we have addressed this by adding relevant references for the importance of FGF signaling (see point 6 in Essential revisions above) (subsection “The non-neuronal epibranchial placodal cell population regulates pharyngeal arch development”). We also tone down the discussion on the signaling centre and clearly state that we do not provide formal evidence that the FGF3^+^ cells form a functional signaling centre (see also the discussion on the role of FGF3 versus other FGFs above).

I therefore feel that while the Eya1-Notch connection is new and very nicely demonstrated, the paper does not move the field forward in a major way by providing new concepts or being highly influential in a broader sense.

We wish to thank the reviewer for the positive words on the Eya1-Notch connection, which we agree is novel, and also represents the first known dephosphorylation of Notch ICD (while there has been a number of phosphorylation events described).

Reviewer #2:[…] Although the data is convincing, the interpretation of the data and model have some weaknesses.1) What is the ligand that is activating Notch1 during the decision between Ngn2-expressing neurogenic placode tissue and Sox2/FGF3-expressing non-neural ectoderm? Dll1 is only expressed in delaminating neuroblasts – well away from the pharyngeal ectoderm – and Jag1 is only expressed in the pouch region. Either there is another Notch ligand responsible for activation of the Hes/Hey genes in the arch ectoderm, or else another signal (such as FGF3 or 8) may be activating these genes. Moreover, Sox2 and FGF3 only label the anterior ("rostral-proximal pharyngeal" as defined by the authors) ectoderm of each arch (Figure 2), yet the Notch target genes Hes1 and Hey1 are expressed throughout the arch ectoderm (Figure 4). Finally, there appears to be very little Notch-ICD staining in the rostral-proximal pharyngeal compared to the arch mesenchyme adjacent to it, both in the wild type embryos and mutants (Figure 4' and D, D'). This may again indicate that other signals might be activating the Hes/Hey genes in this region.

See response to Essential Revision point 2.

2) There is some confusion in the results between whether Eya1 is modifying and regulating the cleaved Notch1 ICD and the intact Notch1 receptor. Figure 4 shows that the amount of Notch C20 staining decreases dramatically in the Eya1 mutants, and the authors conclude this suggests "that the total amount of Notch1 receptor was reduced in the ectodermal cells rather than that receptor cleavage was affected." However, in Figure 6 and Figure 7, the focus is on Eya1 modification and stabilization of the cleaved form. Is it possible that Eya1's threonine phosphatase activity regulates the stability of both the intact Notch1 receptor and its ICD?

See response to Essential Revision point 3.

3) In the rescue experiment in Figure 5, the over-expression of Notch-ICD activates Sox2 in the entire arch ectoderm (Figure 5) but not in the Eya1 mutant (Figure 5). This does not fit in with the model of Eya1-stabilized Notch signaling promoting Sox2 expression in the rostral-proximal pharyngeal ectoderm. The authors need to consider/explain this result, as they don't mention it at all in the text. Part of the problem in interpreting this experiment is that the authors need to show wild type and Eya1^-/-^ embryos as well as the Pax2-Cre; ROSA-N1ICD and Pax2-Cre; ROSA-N1ICD;Eya1^-/-^ images in Figure 5.

See response to Essential Revision point 4.

4) In the Discussion, the authors attribute their arch phenotype to a loss of FGF3: "Therefore, one can assume that phenotypes observed only in Eya1^-/-^ 527 embryos can be attributed to the non-neuronal epibranchial placode, and more specifically to the lack of Fgf3 expression" and: "The failure to organize the PA architecture in the Eya1^-/-^ 547 embryos are likely due to defective FGF signaling". FGF3 mutant mice do not have a clear arch phenotype. Ectoderm-specific FGF8 mutants have an arch phenotype that is much more severe than Eya1 mutants. The authors need to change these parts of the discussion as it impacts their model.

We agree that *Fgf3* is a more unlikely candidate to mediate the effect, for the reasons listed above. As discussed above, we however believe that the upregulation of Sprouty expression by Notch ICD demonstrates that Notch acts upstream of FGF signaling (new Figure 2’). We have modified the text accordingly, and have toned down the discussion of *Fgf3* as a potential morphogen and the *Fgf3*-expressing cell cluster as a potential signaling centre. The model in Figure 8 has also been modified and toned down in accordance with these changes (see also point 5 and 6 under Essential revisions).

In sum, I think the data of the paper is very solid, but at the moment I think their interpretation and model is incomplete.

We appreciate these positive comments, and the model has been amended.